# The Multiomics Analyses of Fecal Matrix and Its Significance to Coeliac Disease Gut Profiling

**DOI:** 10.3390/ijms22041965

**Published:** 2021-02-17

**Authors:** Sheeana Gangadoo, Piumie Rajapaksha Pathirannahalage, Samuel Cheeseman, Yen Thi Hoang Dang, Aaron Elbourne, Daniel Cozzolino, Kay Latham, Vi Khanh Truong, James Chapman

**Affiliations:** 1School of Science, STEM College, RMIT University, Melbourne, VIC 3001, Australia; sheeana.gangadoo@rmit.edu.au (S.G.); s3758115@student.rmit.edu.au (P.R.P.); s3741431@student.rmit.edu.au (S.C.); yenthihoangdang@gmail.com (Y.T.H.D.); aaron.elbourne@rmit.edu.au (A.E.); kay.latham@rmit.edu.au (K.L.); 2Centre for Nutrition and Food Sciences (CNAFS), Queensland Alliance for Agriculture and Food Innovation, The University of Queensland, St Lucia, Brisbane, QLD 4072, Australia; d.cozzolino@uq.edu.au

**Keywords:** gut microbiome, fecal analysis, analytical techniques

## Abstract

Gastrointestinal (GIT) diseases have risen globally in recent years, and early detection of the host’s gut microbiota, typically through fecal material, has become a crucial component for rapid diagnosis of such diseases. Human fecal material is a complex substance composed of undigested macromolecules and particles, and the processing of such matter is a challenge due to the unstable nature of its products and the complexity of the matrix. The identification of these products can be used as an indication for present and future diseases; however, many researchers focus on one variable or marker looking for specific biomarkers of disease. Therefore, the combination of genomics, transcriptomics, proteomics and metabonomics can give a detailed and complete insight into the gut environment. The proper sample collection, sample preparation and accurate analytical methods play a crucial role in generating precise microbial data and hypotheses in gut microbiome research, as well as multivariate data analysis in determining the gut microbiome functionality in regard to diseases. This review summarizes fecal sample protocols involved in profiling coeliac disease.

## 1. Introduction

A “microbiome” is a community of microorganisms, along with their catalogue of genes, inhabiting a particular environment. The human “microbiome” contains 10–100 trillion of a diverse community of bacteria, viruses, archaea and eukaryotic microorganisms that reside in and outside of the human body symbiotic microbial taxa [1,2]. Through evolutionary development, animals and humans have adapted and formed a relationship with microorganisms, symbiotic under normal and healthy conditions, and unfavorable when imbalanced, leading to a dysbiotic state [3,4]. The first use of the term microbiome appears to have been provided by Whipps et al. (1988) as “A convenient ecological framework in which to examine biocontrol systems is that of the microbiome. This may be defined as a characteristic microbial community occupying a reasonably well-defined habitat which has distinct physio-chemical properties. The term thus not only refers to the microorganisms involved but also encompasses their theatre of activity” [5]. Mammals are colonized with microorganisms at birth from their surroundings, including skin contact from other nearby mammals, the mother’s vagina and the initial neighboring contact from the environment [3,6]. These early influences construct a unique ecosystem that has long-lasting effects into adulthood [7]. The first 24 weeks of human life is crucial for the development and establishment of the gut environment and the body’s various systems [8], and, while the gut composition may be shared among family members, its abundance varies from one person to another depending on their diet, lifestyle, genetics and other various environmental factors [9,10]. The gut microbiota can be subdivided into four main groups; Firmicutes and Actinobacteria, which are Gram-positive bacteria and Bacteroidetes and Proteobacteria, which are Gram-negative bacteria [11]. The gastrointestinal tract (GIT) of healthy mammals is populated with over 10^14^ microbes, mainly dominated by Firmicutes and Bacteroidetes with a ratio favoring the former, [1,11] and due to this large ecosystem, humans can be recognized as “superorganisms” [12]. These microorganisms, within a healthy gut, work in mutual synchronization with the host, structuring the body’s most important systems and physiological functions, including; (1) metabolic functions–the breakdown of nondigestible compounds, synthesizing and absorbing metabolites and vitamins, (2) trophic functions–promoting cell differentiation and proliferation within the intestines and (3) protective functions–establishing a relationship with the host immunity, defending against pathogenic invasion and stimulating inflammatory signals [2,3,4,11,13]. The gut microbiome represents a major determinant of human health and diseases and mutually inhabits the different parts of the digestive system [14].

The long-term-dietary pattern of humans affects the structure and function of the gut microbe population [15]. Studies have shown the significant effect diet can have on the gut microbiome composition and abundance, with different diets resulting in different ratios of microbial populations, such as the increased diversity and abundance of microbial populations shown from a Mediterranean diet [16,17]. Scientists have illuminated the role of the gut microbiota in lipid and protein homeostasis, as well as the microbial synthesis of essential metals and vitamins [18]. These studies linking the gut microbiome to the body’s physiological functions can be dated back as far as the 1960s, with early research carried out by scientists such as Dubos, Schedler, Hegstrand and Hine [19]. Dubos (1965) discussed the relationship formed between the host and the microbiome as an “intimate association” between the microorganisms and different parts of the body due to their anatomical distribution in different organs and tissues and their contribution to various physiological needs [20].

A number of foundation studies over the last few decades have shown the link between various disorders and a perturbed gut, including metabolic disorders [21], allergies [22], autoimmune [23] and psychiatric illnesses [24], some of which often emerge in childhood and continue well into adulthood [25,26]. A disturbance to the normal composition and functional ecological niche of the gut microbiota is known as “gut dysbiosis” and can be caused by the introduction of pathogens, bacterial overgrowth and/or antibiotic use, among various other factors, which may reduce the presence of commensal bacteria [27]. The imbalance of the gut microflora increases the relative abundance of detrimental microbes, which possess the ability to disrupt functional tight junctions and reduce the protective barrier and function of the GIT, thus allowing microbes and their toxic agents to translocate through the gut and reaching other parts of the body and inducing inflammation and disease [4,28].

The use of early genomics techniques, including specialized media and anaerobic chambers, allowed for the detection and cultivation of pure cultures only and could neither determine the function of the community nor its relationships of mixed microbial communities [20,29,30]. However, these methods were restrictive in their inability to investigate the communal relationship of the gut microbiome, as well as the identification of unculturable (~90%) bacterial species (spp.) residing in the mammalian gut [31]. It became essential to develop and extend these methods to explore the symbiotic relationship between the host and the gut microbial community [29], as microbes strongly rely on multiple interactions with other species and their metabolic products to grow and thrive, and cannot, therefore, be isolated and cultivated separately. High-throughput methods were developed to side-step the culturing phase and directly identify bacterial species by extracting their DNA and reconstructing the bacterial gene sequences. One prevalent method, ribosomal RNA (rRNA) phylotyping, generates and compares extracted rRNA gene sequences against the extensive database of previously collected gene sequences of the phylotypic marker [29]. The non-requirement of culture is a major advantage of rRNA sequencing, allowing for the rapid identification and diagnosis of specific microbial communities linked to diseases [32].

The qualitative and quantitative analyses of biological matrices, such as feces, urine, saliva, and blood, have made it possible to understand the relationship between the microresidents in the mammalian gut, their products and their involvement with diseases [33]. The focus on biological matrices has allowed for better diagnosis in clinical settings and the chance to treat untreatable diseases. A common example is the autoimmune disorder, coeliac disease (CeD), which is driven by a complex interaction of genetic and environmental factors and is primarily induced by the ingestion of gluten, the main protein component of wheat and other grains [34,35]. The gluten peptides, precisely the alcohol-soluble fraction of gluten (gliadin), trigger and induce an immune response mainly within the connective tissue of the lamina propria. This interaction occurring between the gluten peptides and the antigens cells within the mucosa can lead to inflammation, the activation of lymphocyte T-cell, the release of autoantigens and eventually result in various symptoms and complications, such as malabsorption, intestinal cancers and extraintestinal disorders, such as anemia [34,35,36]. The predisposing genes for contracting coeliac disease have been identified in 95% of diagnosed patients [36], with animal experiments showing the binding activity of gliadin in the small intestine, causing intestinal permeability and the crossing of the toxic protein across the membrane [37].

This paper reviews the state-of-the-art techniques and methods that have been reported from 2015–2020, with respect to the analysis of the human matrices linked to the characterization of the gut microbiome, with specific insight into CeD. This paper will outline the complex link between autoimmune disease and the gut environment, including the microbial composition, transcriptomic, proteomic and metabolomic profiles [38]. Through fecal analysis, specific biomarkers within microbial communities and their byproducts can be evaluated for an improved diagnosis of CeD. The characterization of human gut microbiota requires a robust instrumental number of analytical techniques and statistical analyses to assist researchers in preparing, analyzing and interpreting results from the microbiome, with a high variation of results commonly found within studies. Overall, the paper will focus on method development, which has been comprehensively distilled for researchers to easily and quickly develop methods of analysis and the linked instrumentation to assist researchers in preparing, analyzing and interpreting results from the gut microbiome.

## 2. Methods of Analysis

There are a number of analytes and markers that can be analyzed from a fecal sample, including microorganisms, metabolites, RNA and proteinaceous products. The development of research areas such as metagenomics, transcriptomics, metaproteomics, and metabolomics have allowed researchers to observe the complex communities of the gut microbiome as well as its various interactions with both the host and the microbiome itself. Figure 1 shows a summary of the information available to be extracted and analyzed from a single fecal sample.

### 2.1. Metagenomics

The identification of bacterial species from fecal specimens is comprised of five main crucial steps; (1) sample collection and storage, (2) cell lysis, (3) isolation and extraction of nucleic acids and (4) amplification and sequencing of the DNA gene sequences [39,40]. These five steps will be discussed separately and include high-throughput methods commonly used within the past ten years (2008–2018).

#### 2.1.1. Sample Collection and Storage

The process of storing fecal specimens is crucial for preserving the genomic information from bacterial cells and other cell components and should be carried within 15 min after defecation [41]. It is not recommended to store fecal samples at normal temperature and for long durations of time as this has been shown to negatively affect DNA intensity, induce variabilities between samples and alter the bacterial composition of major phyla [41,42,43], which can affect statistical analyses later in the process. Low temperatures, mainly 4 °C or −20 °C, are also inefficient to preserve important gut biomarkers, such as Firmicutes to Bacteroidetes ratio [44], resulting in an altered microbiota as compared to the fresh equivalent [45]. Fecal samples are more commonly frozen at ultra-low temperatures, −80 °C, or snap-frozen in liquid nitrogen upon collection. Studies show this temperature storing stage affects neither the bacterial composition and diversity nor the short-chain fatty acid production [46], yielding high amounts of DNA and a high purity [47]. The snap-freezing process retains cell integrity within samples, which reduces the prospect of ice crystal formation [45]. The freeze-thawing action when processing frozen fecal material must not be performed more than four times as an increase in Bacteroidetes species can be observed [48].

In challenging conditions where freezing is impracticable, it is possible to use a medium to store fecal specimens up to 5 days, such as RNAlater [49]. RNAlater could be used as an alternative preservation method, with studies showing no significant difference observed to typically freezing samples at −80 °C, while recovering DNA [47,50], and could therefore be used in addition to freezing at ultra-low temperatures [49]. In contrast, few studies have found opposing outcomes with RNAlater, showing a negative impact on bacterial abundance [51], diversity [52], DNA purity and extraction yields [53,54]. Storage using lyophilization also proved to be ineffective, with failure in preserving species from the *Clostridiales* order and additionally reducing bacterial diversity [52]. The immersion of fecal material in chemicals, such as ethanol or potassium dichromate is suggested to be suitable as a preservation method, ensuring storage of fecal materials to be well preserved for up to one month [52,55], though transportation of such solutions may be hazardous if not properly managed. Other preservation methods include the usage of expensive FTA cards or the commercially available OMNIgeneGUT kit [56]. In summary, “the gold standard” for preserving fecal material at the current time is the method of freezing at −80 °C, which can also include a preceding step with snap-freezing using liquid nitrogen.

#### 2.1.2. Lysis of Cells

The extraction of DNA is achieved using a wet weight of fecal sample between 10 and 50 mg to ensure good DNA quality and quantity, with weight not exceeding 200 mg, as this can cause an overloading of the purification matrix and rupture column filters commonly used in DNA extraction kits [57]. Samples are vortexed and centrifuged multiple times to ensure the removal of fecal debris and homogenization of the contents. Hsieh et al. (2016) [58] showed that homogenization could affect the bacterial proportions in the fecal sample, reducing variation within each sample. The appropriate lysis technique of bacterial cells is crucial in recovering the nucleic acids enclosed in the cell and can vary from the diverse amount of species present in the gut, which largely depends on the structure of the cell membrane. Cell lysis is conducted using three different disruption means; mechanical, chemical and enzymatic, with a combination of the three proving to be highly desirable in achieving a higher yield of DNA sequences. This is a crucial step in any cellular extraction involving cells as it allows for the release of the cell contents, including genomic DNA, RNA, metabolites and proteins.

Numerous studies have shown that the incorporation of a bead-beating step, as mechanical lysis means to lyse cells in fecal samples, can lead to an increase in bacterial diversity, high DNA yield and extraction efficiency [59,60,61,62]. Repeated bead-beating showed an increase in the extraction of Gram-positive bacterial species [59,63], including *Bifidobacterium* [62,64], *Faecalibacterium, Lachnospira, Butyrivibrio* and *Eubacterium* genera [59]. Various aspects of the bead-beating step have been extensively optimized, including the speed and duration of bead-beating as well as the choice of bead size. Rigorous mechanical lysis (speed of 6.5 ms^−1^) is preferable in recovering higher amounts of DNA compared to gentle mechanical lysis [65]. Studies report a bead-beating time between 2 and 5 min increased the abundance of bacterial numbers, but degradation and shearing of DNA were observed once duration exceeded >5 min [60,62,64]. Additionally, the use of a repeated, intermittent bead-beating step can be even more advantageous, resulting in a 1.5 to 6-fold increase in DNA yield and diversity [66,67], generating an improved extraction of DNA, in particular from archaeal microbes [59]. The size of the beads is also particularly important in cell lysis, where a smaller diameter of beads, 0.1 mm, is proven to be more efficient larger bead diameters of 0.5 mm and 1.4 mm [68,69,70]. The combination of different-sized beads—3 mm and 0.1 mm glass beads—can additionally enhance uniform sampling [59]. The material type of beads also affects DNA yield and cell solubilization, with glass and zirconia beads showing superiority over ceramic beads [70]. While there have only been few studies reporting the negative impacts of using bead-beating as a cell lysis step, including DNA shearing [61] and no improvement in DNA yield [44], bead-beating is currently the most common and preferable cell lysis technique to extract DNA for fecal samples.

The use of high-frequency energy in the form of sonication to lyse cells is yet another fairly established mechanical lysis method and has the ability to disrupt thick-walled organisms such as *Bacillus anthracis, Mycobacterium tuberculosis* and *Clostridium difficile* in cultured samples [61,68]. Repetto (2013) demonstrated an optimized method for DNA extraction by combining various sonication cycles along with a strong lysis buffer and freeze/thawing to achieving high DNA recovery [71]. Freeze/thawing is also an effective technique to lyse bacterial cells, as shown to significantly improve the lysis of Gram-positive species, such as *Firmicutes* [44,45]. This disruption technique is achieved by subjecting samples to low-level temperatures, typically at either −80 °C or snap-frozen with liquid nitrogen during the storage stages [44,45,56]. The use of additional mechanical lysis steps, such as cryo-milling (the practice of grinding frozen samples) coupled with bead-beating, can also be used prior to subjection to chemical lysis and has been shown to increase the cell numbers as compared to a stand-alone freeze-thawing lysis method [49].

Nowadays, commercial DNA extraction kits are most commonly used in isolating microbial DNA components from fecal and other biological samples, and those kits have combined chemical, mechanical and enzymatic lysis techniques to disrupt microbial cells, as discussed above. These kits employ lysis buffers containing strong detergents, salts, buffers, enzymes, chelating and reducing agents [72,73]. The selection of enzymes effectively with mutanolysin shows increased efficiency in lysing bacterial cells compared to lysozyme, the latter demonstrating low extraction efficiency to archaeal cell wall types [59,74,75,76]. The use of a chemical or an enzymatic lysis stage alone produces inadequate DNA yields low DNA quality [59,62,77]. A summary of common chemicals and enzymes used for lysing bacterial cells in fecal samples in the last ten years, Table 1. Popular commercial kits, including RNeasy PowerMicrobiome kit and PureLink Microbiome DNA purification kit, demonstrate a combination of chemical and enzymes along with physical lysis steps such as cryo-milling, sonication and/or bead beating can result in a greater representation of bacterial diversity, abundance and composition than employing any of the cell lysis techniques on their own [74,78].

#### 2.1.3. Isolation and Extraction of DNA Sequences

The following search criteria, “*DNA extraction in fecal specimens of human origin,*” between 2008 and 2018, was used and yielded a number of key papers, which will be discussed. The extraction and isolation of DNA sequences from fecal samples involve the use of commercial kits—these protocols have been optimized to provide fast and reliable results with little to no preparation using chemicals, making this a preference for large-scale studies (n > 100) [86]. Few studies have used alcohol precipitation as a method for extracting DNA from fecal material; however, this technique involves multiple steps, increased labor, is prone to variation and time-consuming, and was therefore not included in this review.

The most popular commercial DNA extraction kit was found to be Qiagen QIAamp DNA Stool mini kit (QIA-M), which uses an enzymatic approach [55,87,88,89,90,91,92,93,94,95,96,97], followed by MoBio PowerSoil DNA Isolation Kit (MoB-PS) [54,56,98,99,100]. Nechvatal et al. [50] showed that QIA-M combined with RNAlater storage had high DNA recovery, with little PCR inhibition, as compared to MoB-PS. In contrast, MoB-PS demonstrated higher and purer DNA yields than QIA-M [44,101] and a modified version of QIA-M that included a bead-beating step [44].

Comparative studies, however, have shown that neither QIA-M [59,102] nor MoB-PS [102] may be efficient in extracting Gram-positive species, with QIA-M producing DNA extracts of low yield and purity and requiring an RNase treatment and clean up step [101]. The majority of these studies confirmed fast DNA spin kit for soil (FS-S) to be a superior protocol to the two methods mentioned above, producing higher levels of purity and DNA yields, as well as high extraction efficiency, diversity and abundance of the gut microbiota profiles [57,62,64,65,102]. FS-S was effective at extracting high amounts of *Bifidobacterium* spp. [62,64], Lachnospiraceae [102], Actinobacteria [64], *Eubacterium rectale—Blautia coccoides* clostridial group and *Clostridium leptum* group [65] as compared to QIA-M. Other kits displaying better extraction efficacy have included the use of automated extraction using QIAsymphony virus/bacteria midi kit [101] or InviMag Stool DNA kit with a KingFisher magnetic particle processor [103], ZR Fecal DNA miniprep kit [101], NucleoSpin Tissue mini kit [104] or PSP Spin Stool DNA Plus Kit [105]. As mentioned previously, the two most current kits include the PureLink Microbiome DNA Purification and RNeasy PowerMicrobiome kits, proving to be highly effective with a combination of mechanical and enzymatic pretreatment [78].

The assessment of DNA yield can be performed by quantitating the concentration of the extracts on the Qubit 2.0 Fluorimeter (dsDNA HS and ds DNA high-sensitivity assays kit; Invitrogen) [106,107]. Additionally, a NanoDrop spectrophotometer (NanoDrop Technologies, Inc., Wilmington, DE, USA; Isogen Life Science, Utrecht, the Netherlands) can also be used to assess the yield and purity of the extracts at the 260/280 absorbance ratio, where pure DNA will have an OD_260/280_ ranging between 1.7 and 2.0, suggesting that most RNA and protein products have been removed from the extract solution [106]. However, comparisons suggest the Qubit fluorometer has the ability to discern higher concentration extracts than the NanoDrop spectrophotometer [78]. Often agarose gel electrophoresis is used in combination to assess and confirm the integrity of extracted DNA, such as state of degradation and shearing resulting from harsh extraction procedures [106].

#### 2.1.4. Amplification and Sequencing

The next step in analyzing bacterial DNA in metagenomics is the amplification and sequencing of the extracted DNA. Most studies from 2008–2018 utilized polymerase chain reaction (PCR) to amplify bacterial DNA sequences, where specific primers are added targeting the 16S rRNA sequence and identifying the different bacterial strains present in the fecal samples. The presence and abundance of microbial species are purified and visualized on gel electrophoresis prior to PCR amplification [43,45,67,88,98,108]. A mixture containing target DNA, master mix, forward and reverse primers is produced, focusing on the hypervariable V1–V8 regions, particularly the V1–V3 [43,45,67,88,98,108] and the V3–V5 regions [41,62,87,109] as seen in most studies. Alcon-Giner (2017) reported that the regions V4–V5 might overrepresent genus with high G+C content, such as *Lactobacillus*, whereas using V1–V3 regions resulted in a lower diversity of the microbiota composition [62]. Universal primers HDA1-GC and HDA2 may be used in bacterial species isolation for CeD patients, as well as Lac1 and Lac2-GC, representing *Lactobacillus*-related species and primers g-BifidF and g-BifidR-GC representing *Bifidobacterium* spp. [110].

Several apparatus tools can be used for the amplification of PCR products such as targeted DNA, and commonly include the thermal cycler [56,87,88,111] and ABI Prism Fast real-time PCR systems for quantitative PCR (qPCR) [44,91,92,94]. Here, we previously described current PCR techniques and a summarized protocol, including limitations in detecting pathogenic microorganisms [112]. PCR can be coupled with denaturing gradient gel electrophoresis (DGGE) to facilitate the identification of wild and mutated gene sequences [57,113], allowing for separate migrations of these bands as different temperatures are required to denature the differing hydrogen bonds [114,115]. Sequencing is then performed on the cleaned amplicons using next-generation sequencing (NGS) such as Illumina sequencing [56,62,98,116] or pyrosequencing with a 454 GS FLX Titanium sequencer [54,102,108,117].

A popular technique for fast identification and quantitation of specific and/or total bacterial groups in fecal-derived supernatant includes fluorescent in situ hybridization (FISH) in combination with flow cytometry. Available probes, EUB 338 probe, target a conserved region in the domain bacteria, while specific probes, such as Bif 164 and Bac 3030, target *Bifidobacterium* and *Bacteroides/Prevotella* groups. Additionally, the detection of antibody-coated bacterial groups can be detected through the use of fluorescein isothiocyanate-labeled antibodies and quantified through flow cytometry, resulting in the detection of immunoglobulin-coated bacteria associated with gut inflammation and dysbiosis [118,119].

#### 2.1.5. Bacterial Identification

The processing and analysis of the data acquired are crucial for an appropriate and true depiction of the gut microbiome’s community and stability [120]. The bacterial DNA sequences, extracted from fecal samples, are used in differentiating the gene patterns of the gut microbiome [121,122] and can be interpreted and identified using bioinformatic software such as MOTHUR, Greengenes and QIIME 2 [120,123,124,125,126,127]. OTUs are assigned to bacterial DNA sequences based on a value of 97% similarity or higher, with normalization to control for differences in sequencing depths (a given number of reads of a nucleotide in a sequence), diversity (species present and its relative abundance) and log transformations to standardize variances, such as dominant and rare bacterial species [128]. The quality control process is composed of filtering criteria excluding single sequences present from the OTU, sequence reads shorter than 200 bp, homopolymer reads of longer than 8 bases, ambiguous bases and chimeric sequences [125,126]. Prodan, A. et al. 2020 compared multiple bioinformatic pipelines and concluded USEARCH-UNOISE3 performed the best in regard to resolution and specificity of 16S rRNA amplicon sequences, while DADA2, USearch-UPARSE and MOTHUR demonstrated reduced specificity [129].

### 2.2. Transcriptomics

Transcriptomics allows an insight into the functional activity and alterations in bacterial gene expressions in a gut microbial system, by extraction and sequencing RNA products, with a focus on messenger RNAs (mRNAs) [130]. The analysis of these mRNA products allows the understanding of how the genes contained within a microbial community respond differently to various environmental, lifestyle and diet conditions [131] by activating different metabolic pathways [132]. The process typically involves the isolation of mRNA from total RNA products, including rRNA and tRNA, its fractionation and conversion into cDNA sequences [132]. These data then give a functional representation of how the microbial genes are regulated from various stimuli.

#### 2.2.1. Sample Storage and Cell Lysis

The primary focus, when storing fecal specimens for transcriptomics, is to ensure the prolonged stability of RNA products as they degrade at a much quicker rate than DNA [131]. RNA*Later*-ICE medium, storing at −70 °C, can be employed if fecal samples are required to be preserved for long periods of time. This storage method shows considerable stability of RNA molecules [133] for up to 6 days at room temperature, with less variability and mRNA decay, as compared to preservative medium RNA Protect [134]. Different storage conditions can achieve minimal variability in RNA products extracted but can also slightly affect gene regulation; bacterial samples fixed in ethanol can respond to the new carbon source, while bacteria stored in RNA*Later* have displayed osmotic stress due to the high saline content of the preservative solution [135]. Feces can also be stored in phosphate-buffered saline (PBS) [131] or quenched with methanol-HEPES buffer [136] and stored at −80 °C. RNA extracted within 5 h of collection or stored in RNA*Later* displayed greater integrity, with RNA integrity numbers (RIN) of 9 and 7.8, respectively, which was higher than ethanol or the process of freezing on its own [135,137].

The methods employed in lysing bacterial cells for the extraction of RNA are similar to those used to extract DNA species, with chemical, enzymatic, physical or a combination of all three techniques [135,137]. Trizol reagent has been shown to be efficient in lysing bacterial cell contents to extract RNA from cell cultures [138] and from the rumen and mouse fecal samples, in combination with bead beating [139]. Similar to DNA extraction, it is preferable to combine bead-beating with a lysis buffer to ensure maximal RNA extraction, such as the use of beads and Trizol reagent resulting in the production of RNA of higher quality and quantity than generated by enzymatic lysis only or hot sodium dodecyl sulfate/phenol extraction [136,139,140]. Another comparative study [134] showed that the MoBio Powersoil Microbiome kit performed better than the following methods: stool total RNA purification kit (Norgen), lysozyme/mutanolysin pretreatment with the addition of RNeasy mini kit (Qiagen) and an optimized manual extraction by Zoetendal (2006) [141]. The latter method yielded almost double of the RNA quantity than the Powersoil Microbiome kit but had lower RNA quality, and the method was slower and less reproducible. Other kits, including the RiboPure-bacteria kit [131], Qiagen RNA/DNA Midi kit [50] and ZR-Duet DNA/RNA Miniprep kit, can be used in addition to a prior step of lysis buffers and bead-beating combination [140]. Automated systems in extracting RNA products from fecal samples, such as the semi-automatic extraction-NucliSENS easyMag, are also commonly employed [142]. Other automated extraction methods, such as the MagNa pure LC, make use of an automatic mechanism to separate nucleic acids and glass beads to isolate RNA [133], while the BioRobot-EZI employs the lysis buffer and glass beads combination along with the addition of chloroform and isoamyl alcohol solutions [143]. The table below (Table 2) shows the quantity and quality of RNA generated by the most efficient methods of two comparative studies, Kang (2009) [139] and Reck (2015) [134]. A low value for the 260/280 ratio indicates protein contamination, while a low value for the 260/230 ratio represents the contamination of salts and other inhibitory substances in the extract solution. The optimal value for both absorbance ratios is ~2 [134].

#### 2.2.2. Purification and Enrichment of mRNA Product

Following cell lysis, the sample is required to go through multiple purification steps to eliminate unwanted molecules such as DNA, protein molecules and other RNA products. The first step is to remove any DNA contamination in the extracted solution, which can be achieved with kits such as TURBO DNA-*free* [138] or RNase-free DNase I [136,139] purchased from either Roche or Qiagen. The next step is to isolate mRNA, and unlike eukaryotes, prokaryotic mRNA does not have a stable poly(A) tail and therefore cannot be reverse transcribed using established methods for eukaryotic mRNAs. Polyadenylation is one means to isolate microbial mRNA by making use of *Escherichia coli* poly(A) polymerase with MessageAmp II-bacteria kit to generate and add a poly(A) tail to the end tail of the prokaryotic mRNA [131]. Other means of enriching mRNA include the use of probes and magnetic beads that target, attach and remove unwanted rRNA species [138]. Microb*Express* bacterial mRNA enrichment kit (Ambion) [136] and Ribo-Zero kit are the two most popular kits used, with the latter proving more efficient in reducing rRNA products and generating high-quality mRNA species [131,137,138,140].

The quality of RNA is determined by electrophoresis [131,136,139], while the quantity can be measured using the Nanodrop spectrophotometer [131,139]. Some studies additionally check for DNA contamination by looking for the presence of 5S, 16S and 23S rRNA peaks with PCR [136]. Figure 2 below is a summary and combination of the most efficient methods used in extracting mRNA transcripts from fecal specimens.

#### 2.2.3. MicroRNA: A New Method for Shaping the Gut Microbiota

MicroRNA (miRNA) is a class of small, non-coding RNA molecules containing 19–22 nucleotides in length. Since the discovery of miRNA in *Caenorhabditis elegans* [144], thousands of miRNA molecules have been investigated in animals, plants and viruses as key players in the regulation of gene expression networks [145,146,147,148]. These small RNAs have also been found in human stool and identified as potential markers of intestinal malignancy [149,150]. A recent study reported that host fecal miRNA found in two primary sources, intestinal epithelial cells and Hopx-positive cells, can modulate the gut microbiota composition [151]. These miRNAs regulate their targets, bacterial mRNAs, and the host controls the gut microbiota via bacterial mRNA degradation or translational inhibition. Liu et al. [151] found that miR-515-5p and miR-1226-5p stimulate the growth of *Fusobacterium nucleatum* and *E. coli* via modulating their targets, 16S rRNA/23S rRNA and the yegH gene, respectively. The influence of miRNAs on gut microbial growth was also confirmed by other studies, showing that supplying of miR155/let7g can alter the microbiota [152] or the expression of 19 miRNAs were significantly different in intestinal epithelial stem cells (IESC) depending on the microbial status [153]. However, the mechanism of how miRNA enters the bacteria and interfere with the growth of gut microbiota is unclear. The findings could serve as a new direction for controlling the growth of gut microbiota. The method for the identification of fecal miRNA as well as gut microbiota-miRNA interaction has not been well-developed. Liu and colleagues described the main steps used to extract RNA (1) Sample collection and storage, (2) RNA extraction, (3) RNA detection, (4) miRNA identification, (5) miRNA detection and (6) miRNA target prediction [151,154,155,156].

### 2.3. Proteomics

The analysis of protein within fecal material is extremely useful and can communicate three important conditions: inflammation and disease status of the host by the release of leukocytes and epithelial cells, microbial functionality as a result of their protein products and proteolytic activity, and specific signatures and biomarkers of an individual’s GIT [157]. The assessment of fecal biomarkers, particularly calprotectin and lactoferrin, serve as inflammation indicators in the diagnosis of IBD and other GIT disturbances, as well as distinguish mucosal healing between clinical remission and acute flares of ulcerative colitis patients following endoscopic examination [158]. These protein markers are significantly higher in young children from ages 2–9 and older adults from ages 60 and over due to the instability of their gut microbiome [159]. Protein extraction methods are challenging to develop due to their ability to exist in different biological forms [157]. Proteomics studies include qualitative and quantitative of total and/or specific protein biomarkers, with a strong focus on their functional activity, also known as functional proteomics, and involves the observation of the enzymatic activity of proteases. Proteomics studies are carried out by either a “bottom-up” approach involving protein digestion and detection of resulting peptides by mass spectroscopy or a “top-down” approach investigating the changes in protein structure due to posttranslational modifications, proteolysis, degradation and/or isoforms [160]. The various types of proteins analyzed require different types of extraction methods and detection assays, utilizing either gel-based or gel-free methods, which will be discussed in this section. Ruiz et al. (2016) comprehensively reviewed gel-based and gel-free technique strategies in selectively isolating and quantifying protein and peptides in complex matrices [160]. The resultant protein and peptide components extracted from stool samples are either human and/or bacterial origins, and both serve a purpose in understanding changes in the GIT system: intracellular protein and peptide products can be used to link bacterial species to their protein expression and function while determining pathway changes under different conditions, while extracellular protein products can indicate alterations in human health as changes in GIT microbiome are occurring, also serving as important biomarkers in gut diseases [161].

#### 2.3.1. Sample Storage

Fecal material is stored at either −20 °C or −80 °C with no differences observed in the host and microbial protein concentrations and proteolytic activity [157,162,163,164]. Fecal calprotectin (FC) levels showed good stability at −20 °C, even after 6 weeks of storage, but showed significant deterioration at room temperature compared to 4 °C, in particular samples containing FC levels of <100 µg/g fecal sample [162]. The optimal amount of fecal sample weight required varies between extraction methods and typically falls between 100 and 1000 mg of a stool sample, and samples can be homogenized using a Vortex-Genie 2^TM^ to obtain a “non-clumpy” fecal slurry composition prior to extraction [165].

#### 2.3.2. Protein Extraction

The use of a protease inhibitor, such as chloramphenicol, is often added to the fecal sample prior to extraction, preventing the enzymatic breakdown of protein products into small proteins and/or peptides [166,167]. Extraction solutions containing buffers such as PBS and NaN_3_, with the addition of glycerol, can deliver high concentrations of microbial and host protein extracts. The use of glycerol demonstrated stable concentration and protease activity of host protein products of up to three months of storage and stable concentration of microbial proteins of up to one month. However, significant deterioration in protease activity was observed after one week of storage regardless of the buffer used [157]. Other buffers include SDS and dithiothreitol (DTT), which aids in the denaturation of protein components by breaking down protein disulfide bonds and ensure stabilization of the proteins and enzymes [165]. For the extraction of extracellular protein components, such as host proteins and protein biomarkers, there is no need for further lysis and/or extraction, and protein products are filtered, concentrated and analyzed directly using mass spectrometry or immunosorbent assays. Figure 3 below shows a detailed summary and combination of the most efficient methods in extracting host protein and biomarker products from fecal matter.

The extraction of microbial protein products, however, require further cell lysis and is achieved by sonication, followed by high-speed centrifugation to remove cell debris [166,167], and/or the use of bead beating to disrupt cells and release protein components [168]. FastPrep24 remains the most common instrumentation used for bead beating in cell lysis, with studies employing different cycles of bead beating of either short intervals of 30–45 s [157,164] or continuous bead beating for 30 min [165]. A study by Morris et al. (2016) determined a minimum of 3 rounds of bead beating, at a speed of 6 ms^−1^ for 30 s, to achieve the maximal amount of protease activity that could be generated from a fecal sample. However, this was slightly reduced after three rounds [157]. Following the release of intracellular components, unwanted cellular matrices are separated and removed, and protein and peptide components are concentrated by high-speed centrifugation [168] and/or filtration using ultracentrifugation membrane filters [166]. A summarized protocol for extracting microbial protein and peptide products is shown in Figure 4 below.

For the purpose of solely extracting biomarker proteins, such as calprotectin, the use of extraction devices are extremely popular as they are time and money efficient and use a little number of resources as compared to manual extraction methods. The traditional Roche device and new technique EasyExtract are both volume-based extractions, and their efficiency was compared when extracting FC from feces of patients who have Ulcerative Colitis and Crohn’s Disease [163]. The Roche device required a high amount of resources, approximately 100 mg sample weight and 4.9 mL extraction buffer, while Easy Extract only required about 30 mg of fecal material and 1.5 mL of buffer. The EasyExtract additionally involved fewer steps and a short duration of time in obtaining concordant results to the Roche device [163]. FC molecule is able to remain stable for longer periods of time, up to 2.5 months at −20 °C, after extraction with buffer versus prior extraction [162].

#### 2.3.3. Digestion and Peptide Isolation

Following extraction, protein pellets can be isolated with either gel-free or gel-based techniques. A gel-free method includes the use of centrifugation and precipitation with trichloroacetic acid (TCA), followed by ice-cold acetone wash, urea resolubilization, sonication and finally, reduction and alkylation with iodoacetamide to prohibit the formation of disulfide bonds. The protein is further diluted in urea and enzymatically digested twice using trypsin to produce peptide components. The peptides are further cleaned with an acidic salt solution and filtered through a spin column filter prior to liquid chromatography-tandem mass spectrometry (LC–MS/MS) analysis and can further store at −80 °C for future use [165,167].

Gel-based methods can employ the use of one-dimensional polyacrylamide gel electrophoresis (1D-PAGE) or two dimensional-PAGE (2D-PAGE) technique. 1D-PAGE is a simple means to separate and isolate protein components from extracted fecal samples. SDS is most commonly used in 1D-PAGE and works by denaturing and rendering proteins negatively charged, allowing their separation by mass when a current is applied, moving the sample towards a positively charged electrode [169]. An appropriate loading buffer is added to the extracted protein solution, and the mixture is heated at high temperatures, enabling denaturation, followed by separation on PAGE gels [166]. A staining solution, such as Coomassie G-250 or prestained marker, is added to fix, stain and determine the desirable protein products to be analyzed. The gel lanes are sliced at the appropriate mass of selected proteins, and protein components are further reduced and alkylated in-gel, enabling unfolding and cleaving of proteins and preventing the formation of disulfide bonds among proteins, respectively. Gel pieces are then washed to remove the excess staining and buffer solution and tryptically digested, producing peptide components [164,166]. Peptide mixture can either be analyzed or vacuum dried and stored at −20 °C until further use. The resulting dried peptides are reconstituted in formic acid, filtered using glass microfiber filters to remove gel particles, and further purified and concentrated using OMIX tips by Agilent Technologies prior to analysis using LC–MS/MS [164].

The 2D-PAGE technique separates proteins and peptide components by their electrophoretic mobility, based on two unique properties, isoelectric point and molecular mass [170]. Pinto et al. (2017) used 2D-PAGE to identify a complex protein pattern of children’s fecal microbiota, where proteins were first separated based on their isoelectric point using a linear pH 4–7 Immobiline^®^ Dry Strip on an IPGphor apparatus. Second, they separated based on their mass/charge ratio using an Ettan Dalt six apparatus and silver nitrate as a staining solution. The selected areas of the gels were slices and further destained with potassium ferricyanide. The proteins are reduced and alkylated with DTT and iodoacetamide and digested using trypsin. Samples are then acidified using formic acid before LC–MS/MS analysis [168].

#### 2.3.4. Detection

Recent studies investigating protein components and biomarkers extracted from fecal samples have focused on using either immunoassays or mass spectrometry techniques, or a combination of both. A variety of proteomics tools have been provided for the identification of biomarkers and protein-related diseases, including intestinal diseases [171] and extraintestinal diseases, such as Type 1 diabetes [172] and cancers [173].

The concentration of protein and peptide components extracted from fecal samples can be assessed using the bicinchoninic acid assay [157,174] or the Bio-Rad protein assay kit II [168]. Enzymatic activity of proteases can be determined by measuring the release of acid-soluble substance with azocasein assay. The processed sample is added to azocasein solution, incubated at 37 °C over 3 h, and the reaction is terminated by adding trichloroacetic acid. Protein components are precipitated by centrifugation at 12,000× *g* for 5 min, and the supernatant is transferred to a tube containing NaOH and the absorbance, at 442 nm wavelength, is measured [157]. Kits, such as the EnzChek Protease Assay Kit, have been developed to measure protease activity with less extensive manipulation and errors [175].

The majority of studies investigating biomarkers of inflammation, intestinal and extraintestinal diseases use immunosorbent assays, such as enzyme-linked immunosorbent assay (ELISA) [163]. Quantitative ELISA can be used in fecal detection and quantitation of lactoferrin [176], endopeptidase MMP-9 [177], alpha-1-trypsin [175] as well as immunoglobulin calprotectin with EliA calprotectin immunoassay (Thermo Fisher Scientific) [178] and CHI3L1 with human chitinase 3-like Quantikine ELISA kit [179]. The use of three different monoclonal plates used in ELISA kits has been reviewed by quantitatively comparing FC levels, namely Buhlmann, PhiCal v1 and PhiCal v2, indicating the latter to be more desirable with a shorter time and room temperature incubations [162]. PhiCal v2 was comparable in detecting FC levels with Buhlmann plates and showed a higher upper detection limit and greater linearity with FC levels as little as 250 ug/g [162]. Other immunoassays include quantitative immunochromatographic (also known as quantitative lateral flow assay) tests such as the commercial Quantum blue high range (Bühlmann Laboratories) for the measurement of calprotectin [179], immunonephelometry for alpha-1-antitrypsin [180] and immunochemical tests for hemoglobin using the commercial HM-JACKarc analyzer [178]. Lehmann et al. (2014) provided a comprehensive review detailing the various fecal markers, including their potential and limitations in clinical applications of IBD [181].

LC–MS/MS is the preferred and most common method of choice in quantitating protein and peptide products extracted from fecal and other biological samples. Most studies have used an HPLC system coupled with an LTQ Orbitrap Velos MS [167], equipped with a nanospray ionization [165,168]. Some studies use data-dependent nanoLC–electrospray ionization MS/MS on a hybrid linear ion trap-FTICR-MS [166], while others have utilized tandem mass tag (TMT) labeling technique coupled with MS to increase the sensitivity and quantitation of differentially expressed proteins in tissue samples and microbial protein products expressed by a change in the gut environment [174]. The protein and peptide mass spectra obtained from LC–MS/MS is further analyzed and searched using UniProt, a human proteome sequence database [174,182].

### 2.4. Metabonomics

Metabonomics, defined as the quantitative measurements of metabolites, can offer fresh insight into the effects of changing factors in the gut microbiome through the ability to measure and then mathematically model changes in metabolites found in biological matrices, such as feces. Metabonomics dovetails with systems biology, provides an integrated view of biochemistry and investigates the interactions between genes, proteins and metabolites of specific cell types [183,184,185]. The epithelium of the gut colon represents a primary barrier that controls the reabsorption of water and molecules into the body [12,186], such as fatty acids, amino acids, amines and phenolic compounds of different boiling points and physiological properties retained in fecal water [12,187,188]. Fecal water can therefore be used as an important indicator of colon health and the change of environment and lifestyle of the host, and be used alongside fecal matter in the extraction of metabolites for the quantitative and qualitative studies of metabonomics [186,189]. The current analytical methods in extracting metabolic information from the fecal sample include gas chromatography-mass spectrometry (GC–MS), LC–MS and nuclear magnetic resonance (NMR) spectroscopy [12].

#### 2.4.1. GC–MS Analysis

GC–MS is currently the most commonly used instrument for metabonomics studies of fecal samples to characterize the metabolic profile of the gut microbiota [190,191]. Numerous studies demonstrated the superiority of GC–MS over LC–MS and NMR spectroscopy, regarding its reproducibility and sensitivity with separating and detecting metabolites [12,186,192]. GC–MS is highly competent in identifying specific metabolites, such as fatty acids and phenolic compounds in fecal samples [12,191], using electron ionization, allowing intricate fragmentation patterns of various compounds and increasing their accuracy in mass spectral matching with recorded libraries [191]. A suitable fecal metabolite extraction method would typically extract various compounds with high extraction yield [193].

Typically, the first step in processing fecal samples for metabonomics is homogenization by ultracentrifugation at 4 °C with a high-speed of 50,000 rpm for 2 h. Metabolite distortion can occur when preparing large samples of fecal water due to the metabolic alteration occurring from the main gut microorganisms post feces collection. It is therefore recommended that the feces sample is frozen soon after collection [12], and a chemical agent, such as sodium nitrate, is added into the sample as an antimicrobial agent prior to freezing and usage [12,194]. The samples can then be stored for future analyses at 4 °C [186], −80 °C [11,12,195] or additionally lyophilized prior to freezing [195]. Some studies freeze-dry the samples overnight to remove the moisture content, facilitating the mixing with extraction solvents and derivatization reagents [194,196].

Following thawing and vortexing [12], a suitable extraction solvent is added to the fecal water samples, which should rapidly inactivate the metabolism and recover all classes of compounds without altering the chemical or physical state of the metabolites. Many factors can affect the extraction yields of fecal water samples, such as the pH of the aqueous solution. An acidic environment can reduce the solubility of fecal compounds by impacting their ionization, and thus decreasing the extraction efficiency of the solution [12]. This effect can be reversed by freezing the fecal samples, improving the extraction of metabolites and increasing the concentration of amino acids. There was no difference observed with basic solutions when comparing frozen and fresh samples [12]. Commonly used extraction solvents include methanol [11,12,186,197], ice-cold solvent mixture of methanol:water for lyophilized fecal samples [18,33] and deionized water, where the latter has demonstrated high-efficiency in extracting numerous metabolites, including fatty acids and phenolic compounds, as compared to acidified or alkalified water [12,186]. Studies have shown methanol yields the highest amount of components in the extraction [198] as compared to other solvents like cold methanol (−40 °C) and chloroform–methanol–water mixtures of extraction solvents [186]. A significant step in subjecting derivatized samples to GC–MS analysis is to abstain from overloading the extracts within the sample [191], as a high concentration of extract solution can overload the instrument [186], while a lower concentration of the extract can maximize the resolution of GC peaks and increased quantities of metabolites recovered [199]. Additionally, it is shown that larger amounts of materials can reduce the contact between the fecal material and extraction solvent and the overall extraction efficiency [186,200].

Fecal samples can also be subjected to a series of pretreatments prior to derivatization to achieve the maximal extraction of compounds. This process ensures that the compound is separated and eluted effectively, without affecting its stability due to high temperatures often used by GC–MS and other spectrometric instruments [12,201]. The samples are vortexed and centrifuged at the ultra-high-speed [197], with the action of vigorous shaking recovering the molecules of interest while eliminating the protein fractions [11]. The samples are then dried using various techniques: vacuum desiccation, a rotary vacuum concentrator or a gentle stream of nitrogen gas (mixed in toluene kept in anhydrous sodium sulfate) at 50 °C, and further stored at −80 °C prior to the derivatization step [11,33,186,196]. The derivatization process improves the volatility of polar compounds in the fecal sample, increasing their molecular stability, ionization and detectability through the GC–MS [12,194,202]. Chemical derivatization can be achieved using numerous methods, including methoximation using methoxyamine-HCl [33,186,195,196], and/or trimethylsilyl (TMS) silylation reaction with a combination of N-methyl-N-(trimethylsilyl) trifluoroacetamide (MSTFA) and 1% trimethylchlorosilane (TMCS) following incubation [11,33,186]. If desired, an alkane mixture (C_10_–C_32_) can be added to the sample prior to GC–MS analyses to detect and ensure viable reproducibility [186]. The use of ethyl chloroformate (ECF) as the derivatization agent (with an optimized ratio for water:ethanol:pyridine as 12:6:1) can also be used to perform derivatization on human fecal water as it decreases and narrows the boiling point window of derivatives and recorded better peak intensities of derivatives than methyl chloroformate [12]. A study has shown reagents such as pyridine can trigger the derivatization reaction, while alcohols and water have the ability to change the structure of derivatives and affect the solubility of the reaction medium [12]. The reaction speed of derivatization can be accelerated through vortexing and ultrasonicating [12,197], where the vigorous vortexing inhibits the cyclization of reducing sugars and decarboxylation of α-keto acids [197]. Following centrifugation, the derivatized samples are transferred into a small crim top glass vial with anhydrous granular sodium sulfate to remove water traces, and hexane is injected into samples before subjecting them to GC analyses [12,197]. A summarized protocol for GC–MS analysis of human gut metabolites is given in Figure 5.

Yin Ng et al. 2012 optimized an untargeted metabolomic method for analyzing human fecal samples infected with the microbial parasite; *Cryptosporidium*. Optimized extraction conditions detected 135 metabolites from 16 human fecal samples belonging to the classes of amino acids, carbohydrates, organic acids, alcohols, amines, amino ketones, nucleosides, bile acids, fatty acids and indoles in both *Cryptosporidium*-positive and -negative samples. The authors recorded the highest metabolite extraction, approximately 356 components recovered, with methanol solvent at room temperature performing better than cold methanol (−40 °C) or a mixture of chloroform: methanol: water. There was no significant difference recorded between the yields of different components in these solvents [186].

Phua et al. 2013 have developed a method for a gas chromatography time of flight mass spectrometry (GC-TOFMS) method capable of global metabonomic profiling of human feces. A wide range of metabolites was identified, including carbohydrates, carboxylic acids, hydroxyl acids, fatty acid esters, polyols, long-chain alcohols, sterols, phenols, amino acids and nitrogen-containing compounds in human feces. The detected fecal metabolites ranged from endogenous metabolites associated with a host like cholesterol, urea and gut microbes to exogenous xenobiotics. Lyophilized feces detected a wider metabolic space of 704 peaks compared to fecal water with 664 peaks in GC–MS analyses [33].

Gao et al. 2009 demonstrated the intensity and purity of metabolite derivatives to be strongly affected by the use of propanol and n-butanol. Short-chain fatty acids (SCFAs) metabolites were overlapped when methanol was used as the derivatization agent while ethanol increased the peak intensity at room temperature. Additionally, the use of ECF recorded better peak intensities of derivatives than methyl chloroformate. These authors identified approximately 180 metabolites from human fecal samples, including predominantly amine, amino acids, carboxylic acids, fatty acids and phenolic compounds, while the method was unable to derivatize inactive compounds like carbohydrates, alcohol and trimethylamine. Some acids such as formic acid, acetic acid and propionic acid could not be detected due to their low boiling point. However, the authors emphasize that GC–MS analysis is a more suitable method for gut metabolite analyses due to its ability to detect the metabolites, even at low amounts in fecal water.

#### 2.4.2. LC–MS Analysis for Metabolites

LC–MS is an effective and powerful technique in profiling the metabolome, capable of separating and detecting complex specimens with high sensitivity, specificity and resolution and is commonly used for diagnosing intestinal diseases [194,203,204]. The technique used in LC–MS is more flexible in separating compounds with few pre-analytical methods required [205]. Stool samples are immediately collected after defecation and stored at −80 °C until analyses [203,206,207]. The weighted fecal samples are thawed at room temperature and diluted with an organic solvent, such as methanol, to produce a fecal water extract [203,206]. The mixture needs to be vigorously swirled and centrifuged to remove the proteins and precipitates/particulates from the solution [203,206] and is further vortexed and centrifuged again prior to injecting into LC vials [206]. The supernatant is filtered through micropore sized filter membranes to avoid blockage within the LC–MS column system [194,203].

Prior to each injection, the LC–MS valves and syringes are washed twice with 600 µL of a mixture 98:2 ratio of water:methanol solution and 600 µL of water:methanol solution at a 20:80 ratio. The samples can be kept at 4 °C at the autosampler during the analyses [206]. Since a lower temperature is used in LC–MS analysis as compared to GC–MS, sample volatility is not required in LC analysis. Metabolite identification of LC–MS is more time-intensive but is able to extract additional information about the metabolites by providing the metabolite structure depending on their retention time [205]. A summarized protocol for LC–MS analysis of human gut metabolites is given in Figure 6.

Huang et al. 2013 demonstrated the malabsorption and disorders of fatty acids were caused by the changes within the gut microbiome of cirrhotic patients, with the most prominent alteration shown to be a dramatic decrease in fecal bile acids. Bile acids undergo complex deconjugation of bacteria, and the main forms of bile acids in the feces are identified as primary bile and secondary bile acids. The results of LC–MS analysis illustrated that intestinal bacterial overgrowth and fat metabolism are associated with a deficiency of intraluminal bile acids in cirrhotic patients [207].

## 3. CeD Disease Gut Profiling

Coeliac disease is a genetically derived autoimmune disorder, with Class II HLA genotypes DQ2 and DQ8 alleles found in all of the coeliac-diagnosed patients. The risk of disease is influenced by the different combinations of HLA-DQ haplotypes. HLA genes encode antigen-presenting cell (APC) receptors found on T-cells, which regulate the immune system. Recent studies show that differential expression of HLA genes in CeD contributes to T-cell mediated autoimmunity [208,209,210]. The genetic susceptibility presented in CeD patients acts as the first catalysis, followed by the presence and interaction of the antigen, in the case of CeD gliadin peptide, with the GIT mucosal immune system, leading to increased permeability of the gut, triggering inflammation and autoimmunity. Further studies found that non-HLA genes were also a major contributor factor to CeD, as they encode proteins involved in the regulation of intestinal permeability. These polymorphisms among non-HLA genes explain gluten-sensitivity in non-CeD patients [208]. This shows that host–microbiome relationship is the biggest contributor to CeD disease, and this part of the review paper will aim in interconnecting the different contributing aspects of CeD, including the presence of specific bacteria, phenotypic variations from host and gut microbes and the resulting production of proteins and metabolites.

### 3.1. CeD Microbial Profiling

Multiple studies have attempted to profile and differentiate between the gut microbial composition of healthy patients and CeD, including treated (T-CeD) and untreated (Ut-CeD) subjects. Such studies observed a high variation among different population groups, disease states, and experimental design and methods. As with most gut dysbiosis, the ratio of Firmicutes:Bacteroides ratio is lower in CeD as compared to healthy patients [118,211], and is driven by the reduction of Erysipelotrichaceae species and Lachnospiraceae species, and the increase of Barnesiellaceae species and Odoricateriaceae species in adult subjects [212]. The overall diversity of fecal microbiota is increased in CeD patients, as shown in studies involving children subjects [213], and the gut profile is driven by two main bacterial populations: *Lactobacillus* spp. and *Bifidobacterium* spp. The diversity of the *Lactobacillus* community is greatly reduced in Ut-CeD children, with significant decreases in *Lactobacillus curvatus* and *Lactobacillus casei* spp. [213], as compared to healthy children. The abundance of *Lactobacillus* spp. was significantly improved in T-CeD compared to healthy controls [214], while an opposing study reported significant decreases in *Lactobacillus sakei* and total *Lactobacillus* populations in T-CeD compared to Ut-CeD and healthy subjects [215]. These differences could be a result of variations among amplicon sequence variants (ASV), suggesting strain differences in proinflammatory or anti-inflammatory activities of *Lactobacillus* spp. [215]. Similar to *Lactobacillus* spp., the diversity of the *Bifidobacterium* population is greatly reduced in CeD with both Ut-CeD and T-CeD subjects compared to healthy children and adults [118,213,214,216]. Ut-CeD children displayed a significantly lower abundance of *Bifidobacterium longum* [216,217] and a higher abundance of *Bifidobacterium breve* [217], while T-CeD children showed significant decreases in abundance of *Bifidobacterium bifidum*, *B. longum* and *B. breve* than Ut-CeD and healthy subjects [216]. Studies have had opposing results with the abundance of *Bifidobacterium adolescentis* in healthy children [213,216], and as *Lactobacillus* populations above, those differences could also be due to small variations at a strain level. Additionally, the diversity of *Bifidobacterium* spp. varies greatly between children and adult subjects, with studies showing Ut-CeD adults displayed a higher abundance of *Bifidobacterium catenulatum* and *B. bifidum* [110]. A six-month study with a gluten-free diet (GFD) treatment increased *Bifidobacterium* levels compared to the placebo group [218] but significantly reduced the presence of *Lactobacillus* population in CeD children.

Minor populations such as *Akkermansia* and *Dorea* were observed to be in lower abundance in CeD subjects [215], while other bacterial species such as *Leuconostoc* populations, specifically *Leuconostoc mesenteroides* and *Leuconostoc carnosum,* were significantly more prevalent in Ut-CeD children, increasing the overall fecal bacterial diversity [213]. *Bacteroides* populations and *Clostridium leptum* were significantly higher in Ut-CeD and T-CeD than healthy children, while *Staphylococcus* populations and *Escherichia coli* were observed to be significantly higher in Ut-CeD subjects only [214], with T-CeD showing significant reductions in *Staphylococcus* populations as compared to Ut-CeD subjects. Additionally, populations of *Enterococcus* spp. were higher in CeD children [217], as well as Proteobacteria and Bacteroides-Prevotella group proportions, particularly in Ut-CeD than healthy children [118,211]. In adult subjects, bacterial populations of *Candida* spp. (*p* < 0.01) and *Saccharomyces* spp. were significantly higher in CeD patients [219]. Current studies have reported the significant role of *Pseudomonas* bacterial populations in gluten immunogenicity and inflammation in CeD disease; *Pseudomonas fluorescens* and *Pseudomonas aeruginosa* have the ability to mimic and metabolize gliadin peptides, respectively, resulting in the activation of T-cells of CeD patients [209,220]. Those same studies showed the ability of *Lactobacillus* populations to effectively degrade these immunogenic peptides produced by *Pseudomonas* spp. [220]. IgA-coated gut bacteria, possibly associated with increased inflammatory response, were more prevalent in the gut of CeD children and its abundance levels shown to increase with age, further separating the microbial composition from healthy children, with major changes occurring at the age of 5 years old [221]. The phylogenetic investigation of gut microbial communities and the combination of Kyoto Encyclopedia of Genes and Genomes (KEGG) metabolic pathway analysis demonstrated the enrichment of bacterial pathogenic pathways in CeD patients involved with bacterial invasion of epithelial cells, otherwise absent in healthy subjects [221].

### 3.2. CeD mRNA and microRNA Profiling

CeD pathogenesis is an interconnection of the host and microbial genotype, and studies have investigated the expression of both host and microbial genes involved in immune markers and various metabolic pathways. Most of these studies have focused on the differential gene expression profiles of CeD host within the mucosa and intestinal samples. The expression of adaptive immunity markers interleukin (IL), in particular, IL-6 and IL-21, was significantly increased in CeD, while gluten-sensitive patients showed a higher expression of innate immunity marker, Toll-like receptor (TLR) [222]. Bradge (2011) identified eight genes that were highly expressed in CeD patients compared to control, most of them involved in the immunity-mediated inflammatory response and tight membrane junction (TJ) pathways [222,223]. Further studies additionally highlighted specific biomarkers differentiating between no inflammation and low-grade inflammation of CeD patients [224], further reinforcing that the detection of those highly expressed genes could be isolated pre-disease and early-stages of CeD. Recent research also has been focused on non-coding regions, including long non-coding RNAs (lncRNAs), which can hold single-nucleotide polymorphisms (SNPs) associated with increased disease risk. The presence of these SNPs on lncRNAs can further affect the regulation and expression of marker genes, and in the case of CeD, can affect the expression of *IL18RAP* locus involved in the NFkB pathway in the small intestine. CeD susceptibility was also associated with polymorphisms in tight junction genes involved in intestinal integrity [225]. Fecal transcriptomics of CeD gut studies should further be developed to include for the detection of these RNA markers allowing for non-invasive detection of pre- and early CeD diagnosis.

Quantitative trait loci (QTL) analysis has shown the strong influence human genetics and differential expression profiles can have on the gut microbiome environment [226], with a different immune response phenotype expressed by CeD subjects, resulting in altered host defense and immunity against pathogens and toxic products such as deamidated gluten peptides [227]. The analysis of gene expressions, involving the enrichment and pathway analysis of highly expressed genes, could emphasize not only the diagnosis of specific biomarkers to CeD but also specific pathways of the gut environment related to immune response, metabolism and transportation, and intestinal integrity in CeD patients. In addition, the detection of these specific biomarkers and pathways could serve as the diagnosis of CeD pre-disease state [222,228]. While there have been few CeD transcriptomics studies involving the use of fecal matter, fecal host transcriptomics can be used to determine the gene expressions of bacterial species involved in pathogenesis; for example, mRNA expression profiles of immune response and actin-cytoskeleton were enriched with *Clostridium difficile* infection [229]. Additionally, fecal host, transcriptomics can be used to investigate the active functional profiles of various stages of CeD [215,229]. The analysis of fecal matter over blood analysis showed an increased representation of energy metabolisms pathways in CeD subjects and, along with computational functional analysis, fecal transcriptomics can predict functional pathways of pre-CeD subjects [212].

The investigation of miRNA profiling of CeD pathogenesis has been limited to duodenal and serological samples. However, it is possible to utilize past information performed on duodenal biopsies to further extract specific miRNA markers from human feces. Felli, C. et al. (2017) comprehensively compiled a list of miRNAs upregulated and downregulated from past studies. The expression of these miRNAs additionally differ among severity of the disease, age and are strongly associated with intestinal and mucosal damage, as well as an immune response [230].

### 3.3. CeD Protein Biomarkers Profiling

Biomarkers of protein and peptide in nature are used as both diagnoses and to monitor adherence to a gluten-free diet (GFD) in CeD patients. The biomarkers used to diagnose intestinal diseases are unspecific and focused on monitoring the inflammation status of the GIT, and those biomarkers include calprotectin, immunoglobulin A (IgA) and immunoglobulin G (IgG) [231,232,233,234]. Recent studies have uncovered the use of specific biomarkers for the sole purpose of isolating and diagnosing CeD pathogenesis, and include immunogenic gluten peptides (specifically 33-mer gliadin peptide) and deamidated gliadin peptides (DGP), as well as antibodies exclusive to CeD subjects only; anti-gliadin antibody (AGA), anti-endomysial antibody (AEA), anti-tissue transglutaminase (anti-tTG–TG2, TG6, TG3) and the neo-epitopes (modification to the antigenic determinant) tTG-DGP complex [235].

As with most intestinal diseases, fecal calprotectin (FC) levels are significantly higher in Ut-CeD as compared to healthy children, with a significant correlation observed between FC and anti-tTG level [234], while T-CeD on GFD showed a reduction of the biomarker [231,232,233]. The detection of intestinal antibodies associated with CeD again is typically performed via serum samples [222], and research are currently focused on developing less invasive ways in detecting those antibodies and include the use of immunofluorescence analysis and ELISA methods, which are capable of utilizing readily available human biological samples such as fecal matter. One study demonstrates the significant increase of these antibodies, namely AEA, IgA anti-tTG, IgA anti-DGP and IgA anti-actin, in CeD compared to healthy patients [236]. There was a clear association revealed between levels of IgA anti-DGP and levels of gluten immunogenic peptides (GIP), demonstrating the possibility of the former antibody to be used as a biomarker in fecal samples of CeD-diagnosed patients [237]. However, the digestion of these antibodies along the GIT passage could reduce sensitivity and detectability of these biomarkers in stools, and therefore further tests should be performed to evaluate their reproducibility prior to clinical use [238]. Studies currently utilize immunoassays, such as ELISA, and immunochromatographic assays to detect the presence of GIP in fecal samples. These techniques employ monoclonal antibodies (moAbs) of G12, A1 and 33-mer peptide, increasing the sensitivity of detecting these biomarkers using ELISA [239,240], with the reactivity of antibodies shown to be correlated with the potential immunotoxicity of the undigested gluten peptides [241]. ELISA also showed higher recognition of numerous antigenic determinants of gluten peptides, allowing the recognition of simple and complex GIP from various foods, types of gluten components and differing host fermentative degradation [242]. GIP significantly decreased in patients after GFD treatment and could further be used to check diet compliance [243,244], particularly when levels were higher in patients who have been on GFD treatment for a more extended period than patients who have only been on a diet from 1–5 years, indicating non-compliance [245]. The consumption of gluten can also increase the levels of fecal tryptic and glutenasic activity [239]. Verdu and Schuppan 2020 reported that the pathogenesis of coeliac disease might be caused and/or driven by bacteria-derived peptides. Their study report two major peptides derived from *Pseudomonas fluorescens* and their ability in mimicking gliadin epitopes to bind with the crystallized structure of T-cell receptors specific to CeD subjects due to a differently expressed human leukocyte antigen (HLA) locus [209,210]. Biomarkers can also include host and microbe proteases, which are useful in investigating proteolytic and enzymatic activities within a particular environment [157]. The use of fecal proteomics can help researchers understand and optimize symbiotics design and predict utilization pathways in prebiotics [160,246].

### 3.4. CeD Metabolic Profiling

The gut dysbiotic environment of CeD, as discussed above, favors the presence of pathogenic species such as *Pseudomonas* while reducing good gut microbes, *Bifidobacterium* and *Lactobacillus* spp., while also resulting in an increase of toxic compounds and lower levels of beneficial metabolites such as SCFAs. Di Cagno et al. have performed multiple studies investigating a range of volatile organic compounds and amino acids among Ut-CeD, T-CeD and healthy patients to identify specific metabolite markers involving CeD pathogenesis. The studies observed total esters to be significantly reduced, and total ketones significantly increased in CeD. Total alcohols, aldehydes, sulfur compounds and hydrocarbons were significantly increased in Ut-CeD but improved in T-CeD with GFD treatment [247,248]. Total SCFAs were significantly reduced in Ut-CeD patients and improved with GFD treatment, particularly acetic, butyric, pentanoic and isovaleric acids [218]. Isovaleric acid, however, was still significantly lower in T-CeD compared to healthy patients [247,248]. Some studies had opposite results with SCFA levels significantly increased in Ut-CeD and T-CeD compared to healthy children and adults due to the inability of villous atrophy to absorb nutrients in the GIT [110,249]. High gluten intake (30 g per day) increases SCFAs compared to gluten-free diet [239], and butyric levels increased with gliadin consumption in mice studies [250]. CeD-diagnosed children showed the presence of microbiota-derived taurodeoxycholic acid (TDCA), a proinflammatory conjugated bile acid. TDCA is mainly synthesized by *Clostridium* XIVa and *Clostridium* XI; the latter increased in the gut microbiota of CeD patients [221].

CeD diagnosed-patients are put on a strict non-gluten-free diet. However, this therapy has been found to be unresponsive in 30% of patients owing to nonadherence to diet [34]. GFD treatment has been shown to augment the concentrations of acetic, butyric and total SCFA levels [218], with more than one year of treatment found to normalize fecal SCFA in children with CeD [251]. Figure 7shows the bacterial species and gut microbiome profiles for healthy, Ut-CeD and T-CeD subjects. The simulated PCA plots were compiled from the data contained in five publications ^111, 213, 216, 247^. The categorical principal analyses show the diverse gut microbiome profiles of healthy, Ut-CeD and T-CeD patients. The prevalence (%) of *Bifidobacterium* spp. shows that the three subjects are comparative, specifically the T-CeD profile, while the healthy and Ut-CeD profiles were mostly driven by the similar *Bifidobacterium* spp. Specifically, the distance between the species was observed between these two subject profiles, Figure 7A. The distance potentially indicates that the contribution of gluten peptides may produce more favorable environments for *Bifidobacterium* to thrive. Figure 7B includes the compiled work of the above publications along with Collado M 2009 [214], to demonstrate the relationship between the investigated bacterial species and the three different treatment groups, with research mostly focused on the extraction and identification of *Bifidobacterium* and *Lactobacillus* spp. in CeD patients, few publications focused on the role of pathogens belonging to the *Pseudomonas* and *Weisella* genera. Figure 7C shows the expression of genes from intestinal biopsies extracted from Ut-CeD patients investigated in two publications, Bradge 2011 and Bradge 2018, with red and green color indicating the reduced and elevated expression of those genes, respectively [223,224]. Fecal transcriptomics of CeD and other intestinal diseases should be the next step of focus in uncovering the mechanics of the gut environment and its relationship with the host. The monitoring of the gut’s gene profiling and expression appears to be an important asset for early diagnosis as well as a follow-up from treatment strategies.

### 3.5. Chemometrics and Machine Learning (ML)

ML and ML-methods used to address clinical problems in autoimmune disease have been emerging, where personalized care and personalized medicine will begin to shape the future for patients suffering from such diseases. Additionally, patients with multiple autoimmune comorbidities will also benefit from this paradigm of personalized healthcare approaches, which chemometrics, machine learning, or artificial intelligence will realize this goal, and many healthcare systems are already beginning to invest in heavily [252]. The use of biochemical sensors, instruments to measure changes in samples, biochemical signature analysis, and then applying chemometrics is routinely used in the environment, food, and the health sciences [253]. These instruments and sensors produce vast quantities of data. The data revolution where a wealth of clinical data, omic data and high throughput technologies are from [254] will allow for this ability to perform fast diagnosis and therefore improved treatment times. The inclusion of multiple types of omic data combined with ML models potentially re-routes the ability to visualize the complete picture of autoimmune diseases leading to more precise insights. By the combination of sensors (off the shelf), extracting clinical data, and the omic data as singular datasets, these provide very little information, especially without methods for proper interpretation. AI and ML have the ability to spot clinically relevant patterns, which will give rise to the stratification of the patient’s data, which will also provide a route for care, estimation of the diseases, diagnosis, management of the illness, monitoring, treatments, and also responses to these said treatments. We feel that ML and AI will begin to move to new paradigms in autoimmune disease diagnosis and treatments.

## 4. Conclusions

The analysis of human feces can provide valuable insight into the gut health condition of the host and involves the methodical extraction and profiling of the complex biological matrix. Due to discrepancies and high variability among analytical methods and resulting materials processed from these differing methods, there is a high requirement for meticulous sample collection and precise data preparation. This review paper summarizes optimized methods in extracting and processing fecal products, such as DNA, RNA, proteins and various metabolites, in constructing a detailed and informational map of the GIT microbial environment. Additionally, the inclusion of the multivariate analysis of the results is significant in generating this information, and machine learning is the future for precision home medicine tailored for autoimmune disorders such as coeliac disease.

## Figures and Tables

**Figure 1 ijms-22-01965-f001:**
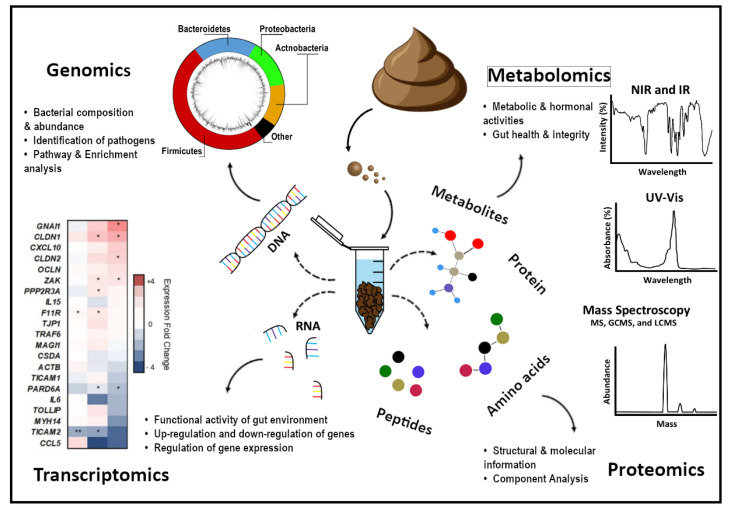
Schematic diagram of information that can be extracted from a fecal sample.

**Figure 2 ijms-22-01965-f002:**
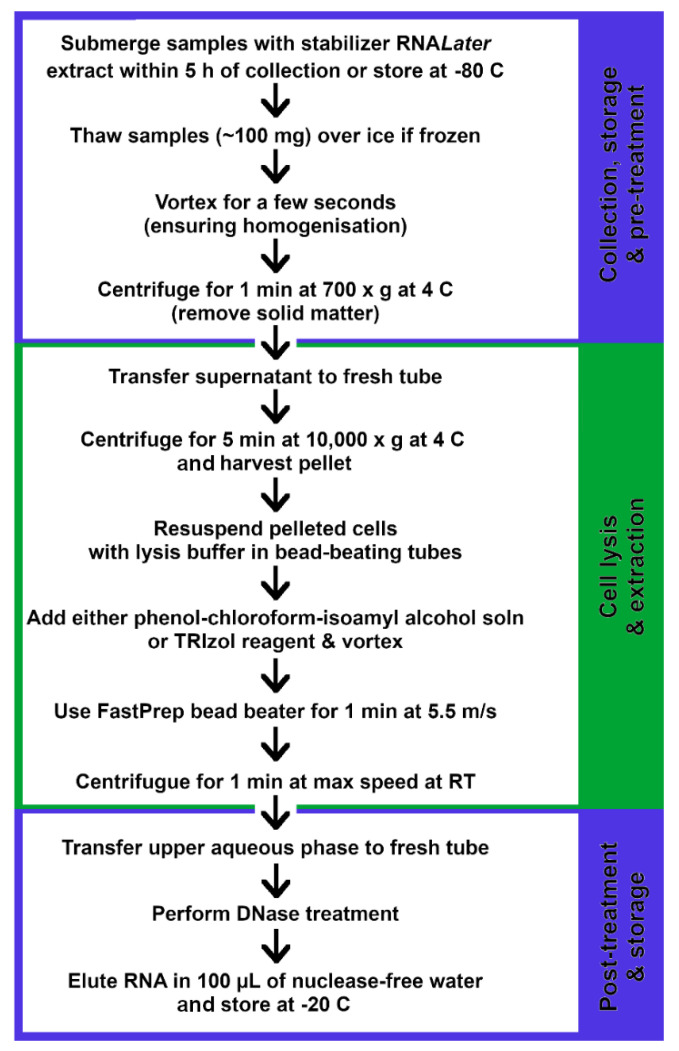
A combination summary of the most efficient methods in extracting mRNA products from fecal specimens.

**Figure 3 ijms-22-01965-f003:**
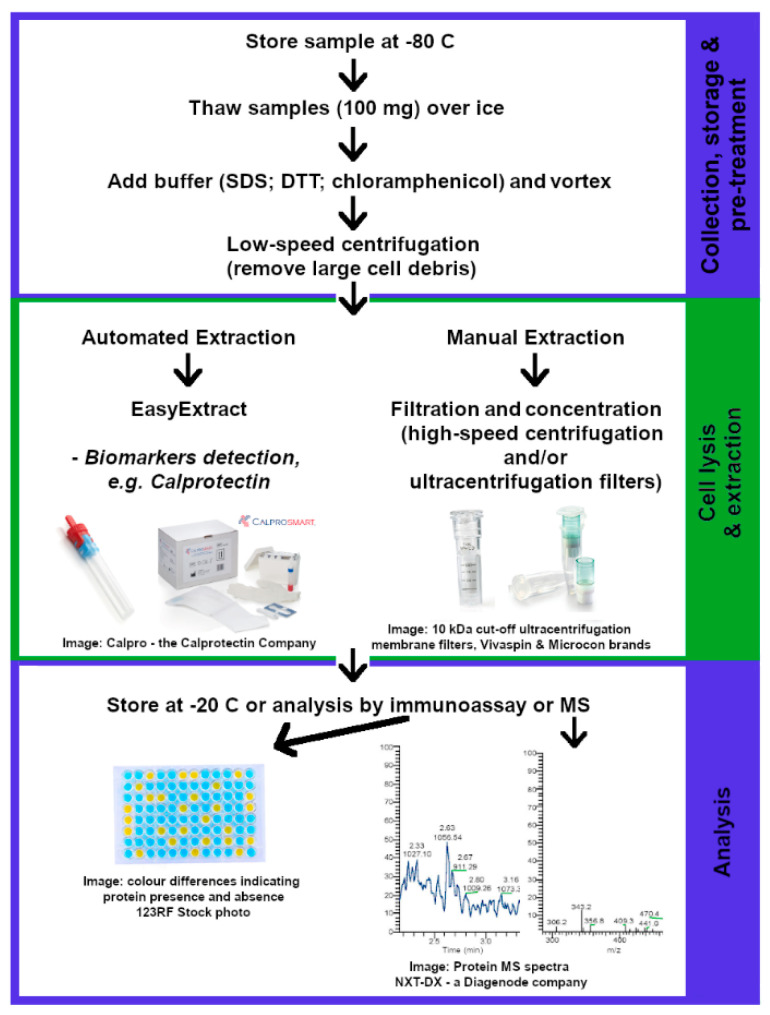
A combination summary of the most efficient methods in extracting host protein and biomarker products from fecal specimens.

**Figure 4 ijms-22-01965-f004:**
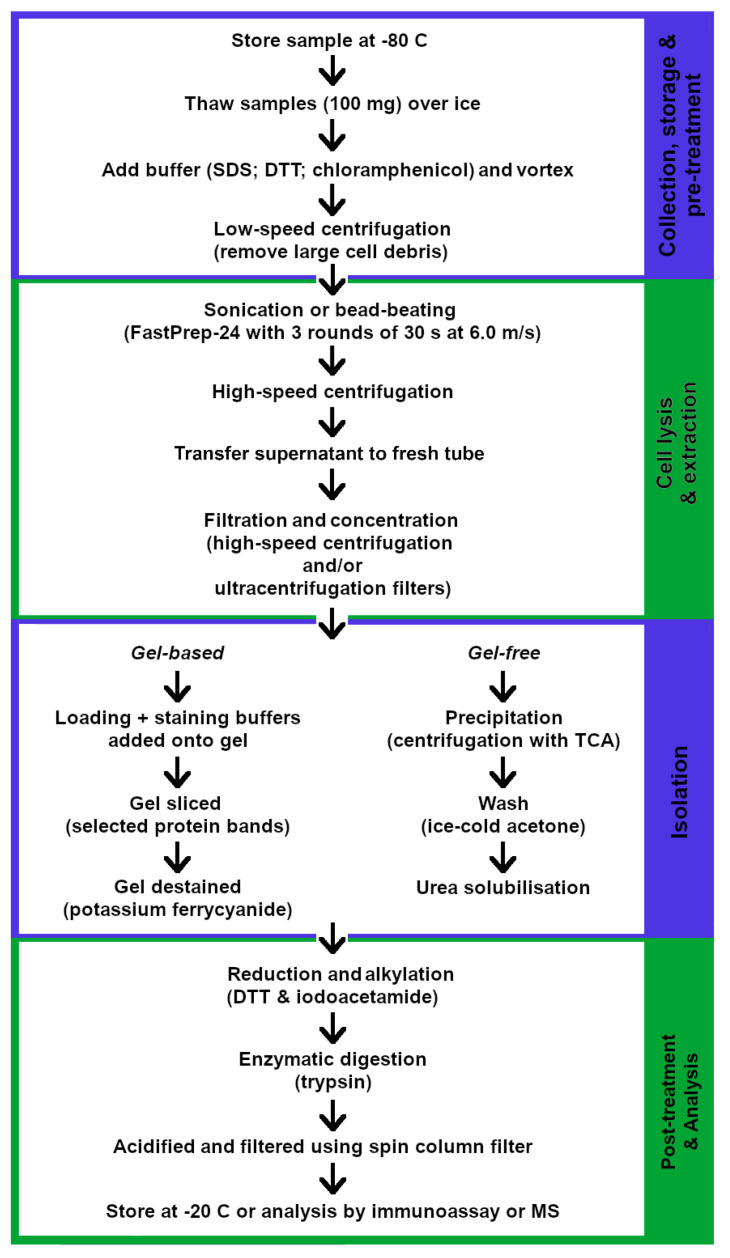
A combination summary of the most efficient methods in extracting microbial protein and peptide components from fecal specimens.

**Figure 5 ijms-22-01965-f005:**
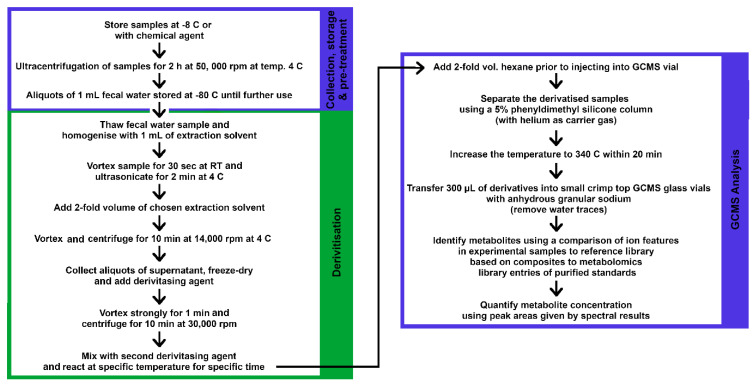
Summarized protocol for GC–MS analysis for human gut metabolites.

**Figure 6 ijms-22-01965-f006:**
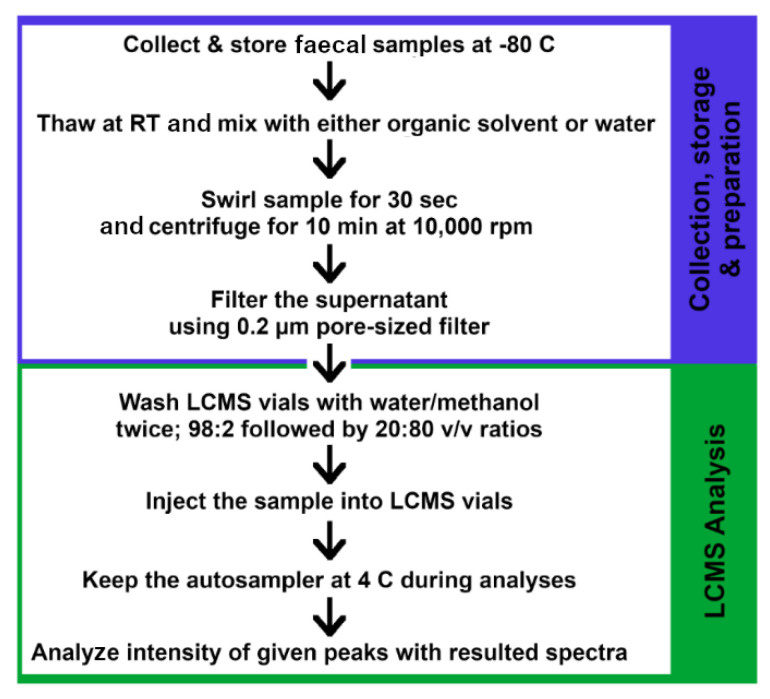
A summarized protocol for LC–MS analysis for human gut metabolites.

**Figure 7 ijms-22-01965-f007:**
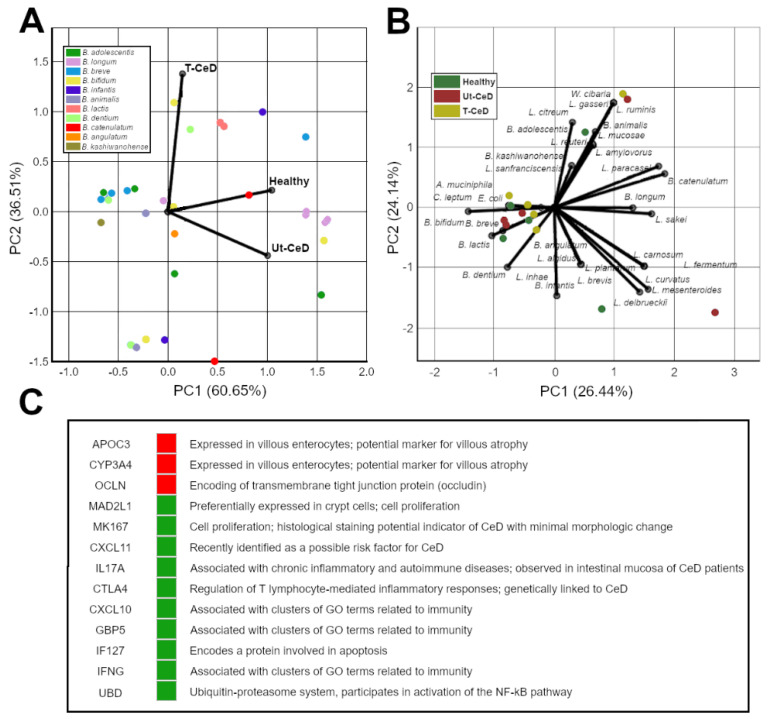
A categorical principal components analysis detailing (**A**) the separation of gut microbiome profiles of healthy, untreated (Ut-CeD) and treated (T-Ced) subjects, (**B**) the influence of specific gut microbial species of those subjects, and (**C**) the profiling of the gut’s gene expression extracted from the intestinal biopsies of coeliac disease (CeD) patients.

**Table 1 ijms-22-01965-t001:** Typical chemical and enzymatic lysing agents.

Enzymes	Mode of Action
Lysozyme	Cleaves the bacterial cell wall [79] by catalyzing the hydrolysis of glycoside-linkages in the peptidoglycan layer [73].
Mutanolysin	Rapidly solubilizes cell walls and removes reducing sugars and free amino groups. Greatly effective against some streptococci strains [80] and Gram-positive cell wall [73].
Guanidine thiocyanate	A chaotropic agent, which disrupts the hydrogen bonding of a molecule and is used to inactivate DNase and RNase, enzymes that digest DNA and RNA, respectively [73,81].
RNAse A	Degrades single-stranded RNA [82]
Proteinase K	Digests proteins by hydrolyzing peptide bonds [83]
Lysostaphin	Specifically breaks some *Staphylococcus* spp. [84] by cleaving the pentaglycine cross bridges of its cell wall [73,85]

**Table 2 ijms-22-01965-t002:** Comparative studies are displaying RNA quality and quantity.

Method	Concentration (μg/g)	Absorbance Ratio 260/280	Absorbance Ratio 260/230
Zoetendal (2006) [134]	>400	<2.0	<1.5
Powersoil microbiome kit (MoBio) [134]	>200	>2.0	>2.25
Bead beating + column [139]	185.1	2.11	2.04
TRIzol Max bacterial RNA kit (Invitrogen) [139]	199.5	2.02	1.03

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
