# Peer review of "The Multiomics Analyses of Fecal Matrix and Its Significance to Coeliac Disease Gut Profiling"

_ijms, 2021, doi:10.3390/ijms22041965_

Round 1

Reviewer 1 Report

D

Dear Authors,

This review is interesting and cover the aspect of analysis related to fecal samples; how to handle the samples and which procedures to follow to evaluate by multi-omics approaches the gut microbiota activity in human. Then after a very long methodological part, the review talks on applications to study Coeliac disease. I suggest at least to revise the review focusing just on this latter aspect.

Unfortunately, the paper is very long, and many terms and definitions are scientifically not correct, thus generating more confusion. NGS are not considered (and this is a big flaw) and paragraph should be included over this aspect. Bacterial identification and quantification are mixed up. Many arguments are repeated more times, and other sentences are redundant and obvious. Moreover, many protocols described are already old, and on the processing of such samples a standardized protocols and procedures already exist (EUFP IHMS). Sometimes the authors, just describe the procedures of commercial kit, giving anything new for literature.

I suggest to accept the paper, but just after major revisions, as:

Focus just on Coeliac disease

Shorten the manuscript

Avoid old references and ambiguous methods

Avoid scholastic talks

avoid talking about blood, saliva, small intestine, general microbiome, but just focous on stools and colon

Include a paragraph about quantification of bacteria and describe qPCR protocols, move here techniques as FISH and flow cyto. Paragraph 2.1.4 is mixing up to distinct aspect as identification and quantification

Include a paragraph just about sequencing of 16S population….(Sanger, Miseq, Hiseq, Nova seq, Pacbio, ARISA, chips,….)…this is fundamental, I wonder why you forgot about it?! Please explain…

Abstract: This section needs clarity; it is very confusing and just appears as an introduction summary. Please, stay close to the meaning of “abstract”

Lines 21-23: The processing of …. with diseases. This sentence is cumbersome, please revise and make it clear.

Introduction:  this section is very long and tedious, it takes a lot of sideways to state a concept. Please, be simpler and clearer. Reviews are made to clarify and make order, not to mix up the fix up.

Lines 37-38: microbiome is colonized…This sentence is ecologically wrong

Line 79: Historical studies…? I do not understand, please be simpler

Line 80: ‘sickly’ gut…non proper term, please change

Line 81: …illnesses, please add some refs

Line 81: rare and emerging diseases… Sorry, but I do not get what are you talking about, please specify

Line 81-82, and indeed into adulthood, please eliminate

Lines 82-85: A disturbance to the…bacteria [21]. There are even other factors generating dysbiosis.

Line 82: please add some refs

Line 91: traditional genomics…I do not understand.

Line 91-93: The use of traditional…..microbial community: this sentence is not clear, please revise

Lines 93-101: Too long and tedious, just to state a concept…please avoid such cumbersome sentences.

Lines 93-96 Specialised media… different species: this sentence is chaotic and needs revision. English must be American and not British…specialized, should be written specialized. Please revise throughout the MS.

Lines 126-142: Please write less to say more….A lot of repetitions of the same

Lines 153-159: I do not like this argument, since it is pretty obvious the difference among the small and the large intestine. Besides, I do not understand the reason to put it at this point. Lastly, if the subject of the review are the stools, why talk about the jejuni? You can arrange it differently if you consider this argument important.

Chapter 2.1: metagenomics for identification of bacteria?? Please revise the concept of metagenomics

2.1.1. Important aspect, but renown. Why don’t you just talk about -80°C and nitrogen

2.1.2 Very long, please shorten. Moreover, it is all down commercial kits. Few scientists still perform home-made methods to sequence the microbiota. The most important thing is reproduciblity and the hassle of pre-treatments could compromise the results. Those days are passed. Nowadays, the Purelink Microbiome DNA extraction kit (Thermo) is the top level.

Figure 2 is redundant

Lines 318-334: Methods…….(2004). Please consider the suggestion reported above. None now could use such methods to publish anywhere.

Lines 342: NanoDrop…Qubit, please add the suppliers.

Lines 343-345: This concept is dated….If DNA is clear and on quantity as suggested by kit instruction, as well as short term stocked, there’s no need to do electrophoresis. Moreover, libraries for NGS amplify small fragments…If you mean other things please specify better,

2.1.4 This section is mixing up different concepts as identification and quantitation. Please revise thoroughly.

Line 362 What!!?? Amplify a PCR product?

2.1.5 MOTHUR is not the only one available, the most of the studies use GreenGenes ad QIME pipeline is more accurate than that you have described..

Lines 396-404: I see a lot of confusion….FISH in this context do not identify, but quantify

Chapter 2.2: Transcriptomics is the extraction…this statement is slippery; extraction and sequencing are more a tool to study transcriptomics. One could even extract and performs “simple” gene expression on a couple of genes, but this is not transcriptomics. Omics are more related to big data from plenty variables. Please revise or comment.

Chapter 2.3: It is well explained, but too long, please cut

Chapter 2.4: It is well explained, but too long.

Chapter 3: This part is different from the previous

Lines 907: Long sentence

Line 911 F/B is not about G+/G-, please correct

Lines 1061-1062: who says so? Please add a ref

Line 1085: spp. please change in species

Lines 1087: Check the citations style

Paragraphs 3.1-3.4: Please avoid including significance values in a review.

Author Response

  • Comment: avoid talking about blood, saliva, small intestine, general microbiome, but just focus on stools and colon

Response: The analysis of blood samples was mentioned from line 101 – 103 to provide an overarching statement on the many types of analyses of various biological matrices, including saliva, to allow for the possibility in determining the gut and its relationship with gastrointestinal diseases, and from line 877 – 880 to highlight the advanced use of faecal samples over blood sample analysis in identifying and predicting gut metabolic pathways of pre-coeliac subjects.

From Line 859 – 862, we have added that “Faecal transcriptomics of CeD gut studies should further be developed to include for the detection of these RNA markers allowing for a non-invasive detection of pre- and early CeD diagnosis.” to give an insight of the current research obtained on coeliac studies performed on intestinal samples that could potentially be translated to faecal analysis. Current faecal gut studies typically focus on by-products resulting from an unhealthy state of the gut microbiome rather than diagnosis of a gut state, which is the direction of research that the authors wish to push towards.

  • Comment: Include a paragraph about quantification of bacteria and describe qPCR protocols, move here techniques as FISH and flow cyto. Paragraph 2.1.4 is mixing up to distinct aspect as identification and quantification

 Response: We had previously described various PCR and flow cytometry techniques and summarised their protocols for the identification of pathogenic microorganisms and the authors have now included this reference in paragraph 2.1.4.

  1. Rajapaksha, P.; Elbourne, A.; Gangadoo, S.; Brown, R.; Cozzolino, D.; Chapman, J. J. A., A review of methods for the detection of pathogenic microorganisms. 2019, 144, (2), 396-411.

 Comment: Include a paragraph just about sequencing of 16S population….(Sanger, Miseq, Hiseq, Nova seq, Pacbio, ARISA, chips,….)…this is fundamental, I wonder why you forgot about it?! Please explain…

 Response: For succinctness, only NGS Illumina sequencing and pyrosequencing were included based on their popularity in the last 10 years, as indicated from lines 317 – 319. The review paper focused on faecal extraction techniques and methods rather than the processing of the outputs obtained from the faecal samples. We feel this is the important part of our review as it gives researchers the opportunity to develop their extraction techniques rather than processing.

  • Comment: Abstract: This section needs clarity; it is very confusing and just appears as an introduction summary. Please, stay close to the meaning of “abstract”

 Response: Abstract was amended. Please see for the overhaul below

“Globally, gastrointestinal (GIT) diseases have risen in recent years and early detection of the host’s gut microbiota, typically through faecal material, has become a crucial component for rapid diagnosis of such diseases. Human faecal material is a complex substance composed of undigested macromolecules, particles, and a plethora of biochemicals, thus the processing of such matter is a challenge, due to the unstable nature of its products, and complexity of the matrix. The identification of these products can be used as an indication for present and future diseases; however, many researchers focus on one variable or marker looking for specific biomarkers of a disease, therefore the combination of genomics, transcriptomics, proteomics, and metabonomics can give a detailed and complete insight of the gut environment. The proper sample collection, sample preparation and accurate analytical methods play a crucial role in generating precise microbial data and hypotheses in gut microbiome research, as well as multivariate data analysis in determining the gut microbiome functionality in regard to diseases. This review summarizes faecal sample protocols involved in profiling Coeliac Disease.”

  • Comment: Lines 21-23: The processing of …. with diseases. This sentence is cumbersome, please revise and make it clear.

 Response: Changed this sentence to “The processing of faecal samples requires essential steps including acquiring the proper analytical tool/method as well as multivariate data analysis in determining the gut microbiome functionality in regard to diseases.”

  • Comment: Introduction: this section is very long and tedious, it takes a lot of sideways to state a concept. Please, be simpler and clearer. Reviews are made to clarify and make order, not to mix up the fix up.

 Response: Sentences were merged and made clearer to the point within the Introduction section, specifically lines 26 – 29, line 37, lines 61 – 63, line 72 – 78, lines 84 – 94, 118 – 128.

  • Comment: Lines 37-38: microbiome is colonized…This sentence is ecologically wrong

 Response: Sentence was changed to The human ‘microbiome’ contains 10-100 trillion of a diverse community of bacteria, viruses, archaea and eukaryotic microorganisms that reside in and outside of the human body symbiotic microbial taxa [1, 2].”

  1. Ursell, L. K.; Metcalf, J. L.; Parfrey, L. W.; Knight, R., Defining the Human Microbiome. Nutrition Reviews 2012, 70, (1), S38-S44.
  2. Shreiner, A. B.; Kao, J. Y.; Young, V. B., The gut microbiome in health and in disease. Current Opinion in Gastroenterology 2015, 31, (1), 69–75.

 Comment: Line 79: Historical studies…? I do not understand, please be simpler

 Response: Historical studies” was changed to “A number of foundation studies"

Comment: Line 80: ‘sickly’ gut…non proper term, please change

 Response: Changed to ‘perturbed’

Comment: Line 81: …illnesses, please add some refs

Response: The following references were added:

  1. Hur, K. Y.; Lee, M.-S. J. D.; journal, m., Gut microbiota and metabolic disorders. 2015, 39, (3), 198.
  2. Perrier, C.; Corthesy, B. J. C.; Allergy, E., Gut permeability and food allergies. 2011, 41, (1), 20-28.
  3. Sprouse, M. L.; Bates, N. A.; Felix, K. M.; Wu, H. J. J. J. I., Impact of gut microbiota on gut‐distal autoimmunity: a focus on T cells. 2019, 156, (4), 305-318.
  4. Rogers, G.; Keating, D.; Young, R.; Wong, M.; Licinio, J.; Wesselingh, S. J. M. p., From gut dysbiosis to altered brain function and mental illness: mechanisms and pathways. 2016, 21, (6), 738-748.

Comment: Line 81: rare and emerging diseases… Sorry, but I do not get what are you talking about, please specify

Response: This sentence was combined and changed to “A number of foundation studies over the last few decades have shown the link between various disorders and a perturbed gut, including metabolic disorders [21], allergies [22], autoimmune [23] and psychiatric illnesses [24], some of which often emerge in childhood and continue well into adulthood [25, 26].”

Comment: Line 81-82, and indeed into adulthood, please eliminate

Response: Changed to “that are well passed into adulthood”

Comment: Lines 82-85: A disturbance to the…bacteria [21]. There are even other factors generating dysbiosis.

Response: Added “among various other factors”

Comment: Line 82: please add some refs

Response: The following references were added:

  1. Putignani, L.; Del Chierico, F.; Vernocchi, P.; Cicala, M.; Cucchiara, S.; Dallapiccola, B.; diseases, D. S. G. J. I. b., Gut Microbiota dysbiosis as risk and premorbid factors of IBD and IBS along the childhood–Adulthood transition. 2016, 22, (2), 487-504.
  2. Munyaka, P. M.; Khafipour, E.; Ghia, J.-E. J. F. i. p., External influence of early childhood establishment of gut microbiota and subsequent health implications. 2014, 2, 109.

Comment: Line 91: traditional genomics…I do not understand & Comment: Line 91-93: The use of traditional…..microbial community: this sentence is not clear, please revise and

Response: Changed to clarify “early genomics techniques”

Comment: Lines 93-101: Too long and tedious, just to state a concept…please avoid such cumbersome sentences.

Response: This paragraph was shortened and simplified to communicate the concept clearly. However, the authors kept most of the original information in the paragraph as they believe it necessary to provide a quick timeline of the adaptation of high-throughput methods to the audience.

“The use of early genomics techniques, including specialised media and anaerobic chambers, allowed for the detection and cultivation of pure cultures only and could neither determine the function of the community nor its relationships of mixed microbial communities [20, 23, 24]. However these methods were restrictive in their inability to investigate the communal relationship of the gut microbiome, as well as the identification of unculturable (~90%) bacterial species residing in the mammalian gut [25]. It became essential to develop and extend these methods to explore the symbiotic relationship between the host and the gut microbial community [23], as microbes strongly rely on multiple interactions with other species and their metabolic products to grow and thrive, and cannot therefore be isolated and cultivated separately. High-throughput methods were developed to side-step the culturing phase and directly identify bacterial species by extracting their DNA and reconstructing the bacterial gene sequences.”

Comment: Lines 93-96 Specialised media… different species: this sentence is chaotic and needs revision. English must be American and not British…specialized, should be written specialized. Please revise throughout the MS.

Response: This sentence was corrected. See response to previous comment above.

Comment: Lines 126-142: Please write less to say more….A lot of repetitions of the same

Response: We thank the reviewer for picking up on this and have removed the repetition to:

“This paper will outline the complex link between the autoimmune disease and the gut environment, including the microbial composition, transcriptomic, proteomic and metabolomic profiles [32]. Through faecal analysis, specific biomarkers within microbial communities and their by-products can be evaluated for an improved diagnosis of CeD. The characterisation of human gut microbiota requires a robust instrumental number of analytical techniques and statistical analyses to assist researchers in preparing, analysing and interpreting results from the microbiome, with a high variation of results commonly found within studies. Overall, the paper will focus on method development, which has been comprehensively distilled for researchers to easily and quickly develop methods of analysis and the linked instrumentation to assist researchers in preparing, analysing and interpreting results from the gut microbiome.”

  • Comment: Lines 153-159: I do not like this argument, since it is pretty obvious the difference between the small and the large intestine. Besides, I do not understand the reason to put it at this point. Lastly, if the subject of the review are the stools, why talk about the jejuni? You can arrange it differently if you consider this argument important.

Response: Reviewer 1 is right. This paragraph was removed as the argument was invalid in the context of the review.

  • Comment: Chapter 2.1: metagenomics for identification of bacteria?? Please revise the concept of metagenomics

Response: As per references below, Metagenomics is a well-used term to describe the study of genetic material extracted from environmental samples including the gut microbiome and the taxonomically classification of uncultivated microorganisms. These references were added in line 146.

  1. Grieb, A.; Bowers, R. M.; Oggerin, M.; Goudeau, D.; Lee, J.; Malmstrom, R. R.; Woyke, T.; Fuchs, B. M., A pipeline for targeted metagenomics of environmental bacteria. Microbiome 2020, 8, (1), 1-17.
  2. Mandal, R. S.; Saha, S.; Das, S., Metagenomic surveys of gut microbiota. Genomics, Proteomics & Bioinformatics 2015, 13, (3), 148-158.

  • Comment:1.1. Important aspect, but renown. Why don’t you just talk about -80°C and nitrogen

Response: The use of -80oC and nitrogen was discussed from lines 158 – 165, with other methods discussed and their limitations explained in detail. A final statement was included, concluding the storage at -80oC to be the gold standard for faecal material preservation from lines 179 – 182.

  • Comment:1.2 Very long, please shorten. Moreover, it is all down commercial kits. Few scientists still perform home-made methods to sequence the microbiota. The most important thing is reproducibility and the hassle of pre-treatments could compromise the results. Those days are passed. Nowadays, the Purelink Microbiome DNA extraction kit (Thermo) is the top level.

Response: This section was greatly shortened, and omitted the use of old techniques, and rather focused on the most advanced and current methods in extracting DNA from stool samples. Additionally, the authors added the use of popular PureLink Microbiome DNA Purification Kit and RNeasy PowerMicrobiome kit.

  • Comment: Figure 2 is redundant

 Response: As instructed we have removed Figure 2

  • Comment: Lines 318-334: Methods…….(2004). Please consider the suggestion reported above. None now could use such methods to publish anywhere.

 Response: Reviewer 1 is right, and this whole section was removed as no current publications used those methods. As a result, figure 3 was also removed.

  • Comment: Lines 342: NanoDrop…Qubit, please add the suppliers.

 Response: Suppliers were added.

  • Comment: Lines 343-345: This concept is dated….If DNA is clear and on quantity as suggested by kit instruction, as well as short term stocked, there’s no need to do electrophoresis. Moreover, libraries for NGS amplify small fragments…If you mean other things please specify better,

 Response: Few studies still use gel electrophoresis in combination with NanoDrop spectrophotometer in confirming the integrity of DNA products extracted from biological samples. As some mechanical lysis can be quite harsh on the DNA product, evidence of shearing and degradation is often visualised using gel electrophoresis.

  • Comment: 1.4 This section is mixing up different concepts as identification and quantitation. Please revise thoroughly.

 Response: This section does not mention the quantitation of PCR products, apart from the brief mention of ‘ABI Prism Fast real-time PCR systems” used for quantitative PCR. However the sentence was changed to induce clarity to “Several apparatus tools can be used for the amplification of PCR products such as targeted DNA, and commonly include the Thermal Cycler [56, 87, 88, 112] and ABI Prism Fast real-time PCR systems for quantitative PCR (qPCR) [44, 91, 92, 94]. Here, we previously described current PCR techniques and a summarised protocol, including limitations in detecting pathogenic microorganisms [113]."

  • Comment: Line 362 What!!?? Amplify a PCR product?

 Response: We have changed line 362 as seen in the previous comment, which is now line 309.

  • Comment: 1.5 MOTHUR is not the only one available, the most of the studies use GreenGenes ad QIME pipeline is more accurate than that you have described..

 Response: Authors included the use of GreenGenes and QIIME 2, as well as a study comparing multiple bioinformatic pipelines:

  1. Hoang, H. T.; Le, D. H.; Le, T. T. H.; Nguyen, T. T. N.; Chu, H. H.; Nguyen, N. T. J. D. i. b., Metagenomic 16S rDNA amplicon data of microbial diversity of guts in Vietnamese humans with type 2 diabetes and nondiabetic adults. 2021, 34, 106690.
  2. Estaki, M.; Jiang, L.; Bokulich, N. A.; McDonald, D.; González, A.; Kosciolek, T.; Martino, C.; Zhu, Q.; Birmingham, A.; Vázquez‐Baeza, Y. J. C. p. i. b., QIIME 2 Enables Comprehensive End‐to‐End Analysis of Diverse Microbiome Data and Comparative Studies with Publicly Available Data. 2020, 70, (1), e100.
  3. Prodan, A.; Tremaroli, V.; Brolin, H.; Zwinderman, A. H.; Nieuwdorp, M.; Levin, E. J. P. O., Comparing bioinformatic pipelines for microbial 16S rRNA amplicon sequencing. 2020, 15, (1), e0227434.

  • Comment: Lines 396-404: I see a lot of confusion….FISH in this context do not identify, but quantify

 Response: Reviewer 1 is right. Paragraph was changed to “A popular technique for fast identification and quantitation of specific and/or total bacterial groups in faecal-derived supernatant include the use of Fluorescent in situ hybridization in combination with flow cytometry. General probes, EUB 338 probe, target a conserved region in the domain bacteria, while specific probes, such as Bif 164 and Bac 3030, target Bifidobacterium and Bacteroides/Prevotella groups.”

  • Comment: Chapter 2.2: Transcriptomics is the extraction…this statement is slippery; extraction and sequencing are more a tool to study transcriptomics. One could even extract and performs “simple” gene expression on a couple of genes, but this is not transcriptomics. Omics are more related to big data from plenty variables. Please revise or comment.

 Response: Sentence was changed to “Transcriptomics allows an insight into the functional activity and alterations in bacterial gene expressions in a gut microbial system, by extraction and sequencing RNA products, with a focus on messenger RNAs (mRNAs)”.

  • Comment: Chapter 2.3: It is well explained, but too long, please cut.

Response: This section was cut, however the authors kept most of the original writing as they believe it to be a significant input in developing and diagnosing specific biomarkers for GIT diseases.

  • Comment: Chapter 2.4: It is well explained, but too long.

Response: This section was refined and now includes only studies involving human subjects and diseases originating from the GIT tract.

  • Comment: Chapter 3: This part is different from the previous

Response: The review paper has two distinct parts to its structure: 1) summary of faecal analysis methods and techniques in assessing the human gut environment and integrity and 2) summary of current celiac studies where such high-throughput techniques have been used, in an attempt to project the ability of those techniques in depicting a close to the accurate representation of the gut from faecal analysis only, as well as predicting disease state. This statement was disclosed from lines 119 – 132.

  • Comment: Lines 907: Long sentence

 Response: The sentence was changed to “Multiple studies have attempted to profile and differentiate between the gut microbial composition of healthy patients and CeD, including treated (T-CeD) and untreated (Ut-CeD) subjects. Such studies observed a high variation among different population groups, disease states, and experimental design and methods.”

  • Comment: Line 911 F/B is not about G+/G-, please correct

Response: This was corrected, and ratio G+/G- was removed to only include F/B ratio.

  • Comment: Lines 1061-1062: who says so? Please add a ref

Response: This statement was concluded based on reviewing current publications on celiac disease, and therefore the sentence was changed to “The gut dysbiotic environment of CeD, as discussed above, favors the presence of pathogenic spp. such as Pseudomonas while reducing good gut microbes, Bifidobacterium and Lactobacillus spp., while also resulting in an increase of toxic compounds and lower levels of beneficial metabolites such as SCFAs.”

  • Comment: Line 1085: spp. please change in species

Response: ‘spp.’ was kept in the manuscript as an abbreviation to species, as added in line 95.

  • Comment: Lines 1087: Check the citations style

Response: We have performed these final checks...

  • Comment: Paragraphs 3.1-3.4: Please avoid including significance values in a review.

Response: Significance values were removed.

Reviewer 2 Report

Reviesed paper "Multiomics analysis of faecal matrix and its significance to Coeliac Disease gut profiling" is very interesting and very important.

All Tables and Figures are perfect. Only quality of Figure 9 is poor - needs correction.

In Review type papers very important is a meta-analysis. In this paper lack of meta-analysis.

Paper needs major revision.

Author Response

Overall comments:

Revised paper "Multiomics analysis of faecal matrix and its significance to Coeliac Disease gut profiling" is very interesting and very important.

Comments requiring changes:

  • Comment: All Tables and Figures are perfect. Only quality of Figure 9 is poor - needs correction.

Response: Quality of figure 9 was improved

  • Comment: In Review type papers very important is a meta-analysis. In this paper lack of meta-analysis.

Response: Data and results of Coeliac research are reported differently across studies, hence affecting the ability for the authors to conduct a meta-analysis. A categorical principal analysis was instead conducted with five publications to highlight the separation of the gut microbiome profiles from coeliac patients to healthy adults, as well as the influence of specific gut microbial species involved with coeliac.

Round 2

Reviewer 1 Report

Dear authors,

thank you for the new version of the manuscript, by my side it is now ready for publication.

Reviewer 2 Report

I recommend this manuscript to publication in present form.